# Multidimensional analysis of screening results of deafness susceptibility genes in 3066 newborns of different altitudes and nationalities in Xining, Qinghai(ISRCTN89197487)

Benhong Ren[1,2,3], Yazhen Wu[4], Wenyuan Gan[1,2], Qingping Zhang[1,2], Dandan Yang[4], Wenjun Cao[1], Xiaoli Zhang[1], Yuan Jiang[5], Ying Zhang[1], Bin Guo[1], Yongxia Tie[6], Jiannan Liu[7], Guilan Cai[8], Yi Wang ![ORCID][1,9]*, Yanyan Ma[10]*

**1** Otolaryngology Department, Qinghai University Affiliated Hospital, Xining, Qinghai Province, China, **2** Otolaryngology Major, Clinical Medical College, Qinghai University, Xining City, Qinghai Province, China, **3** Department of Otolaryngology-Head and Neck Surgery, Shanxi Fenyang Hospital, Fenyang City, Lvliang City, Shanxi Province, China, **4** Department of Neonatal Intensive Care Medicine, Affiliated Hospital of Qinghai University, Xining, Qinghai Province, China, **5** Central Laboratory, Affiliated Hospital of Qinghai University, Xining City, Qinghai Province, China, **6** People's Hospital of Minhe Hui and Tu Autonomous County, Haidong City, Qinghai Province, China, **7** Qinghai Maternal and Child Health-Care Hospital, Xining City, Qinghai Province, China, **8** People's Hospital of Datong County, Datong, Qinghai Province, China, **9** Department of Otolaryngology, Peking Union Medical College Hospital, Chinese Academy of Medical Sciences & Peking Union Medical College, Beijing, China, **10** Department of Pediatrics, Affiliated Hospital of Qinghai University, Xining, Qinghai Province, China

* wegreatgroup@163.com (YW); mayanyan_research@qhu.edu.cn (YM)

## Abstract

### Objective

In Xining City, Qinghai Province, which is located in a high-altitude hypoxic environment, this study employs high-throughput sequencing technology to conduct large-scale deafness-related gene screening among newborns. It aims to assess the carrier rate and variant types of deafness-related genes in local newborns; and based on different genotypic characteristics, provide targeted health education and clinical guidance to the parents of children with deafness, with the goal of achieving early diagnosis, early prevention, and early intervention of deafness. At the same time, this study also provides scientific basic data support for deafness prevention and treatment-related research in Qinghai Province.

### Methods

From March 2023 to March 2025, the research team of the Affiliated Hospital of Qinghai University strictly abided by ethical norms. With the full informed consent of the participants, the team systematically and continuously collected heel blood samples from multi-ethnic newborns living in areas at different altitudes, and successfully obtained 3,615 complete blood samples required for the study. Subsequently,

**Data availability statement:** Due to the requirements of relevant local regulatory policies in Qinghai Province, the relevant data of this study cannot be shared publicly for the time being. Researchers who meet the eligibility criteria for accessing confidential data may apply for access to the data from the Research Ethics Committee of the Affiliated Hospital of Qinghai University (contact number: 0971-6162033).

**Funding:** This study was supported by the Foundation of Qinghai Provincial Department of Science and Technology of China (Project No. 2023-SF-129).

**Competing interests:** The authors have declared that no competing interests exist.

high-throughput sequencing technology was applied to detect these samples, and accurate data on deafness susceptibility-related genes were obtained. During the analysis phase, the research team took altitude (low altitude, medium altitude, high altitude), ethnicity (Tibetan, Hui, Han, Salar, Tu, etc.), and genotype (15 loci of 4 common deafness-causing genes: GJB2, SLC26A4, mitochondrial 12SrRNA, and GJB3) as the core dimensions. By comprehensively using statistical analysis and bioinformatics methods, a systematic multi-dimensional analysis was conducted to deeply explore the association characteristics between different factors and the carriage of deafness susceptibility genes.

## Results

This study collected a total of 3,615 cases of newborn blood spot specimens. After excluding 549 cases of substandard specimens, a valid screening cohort of 3,066 newborns was eventually formed. Within this valid cohort, 165 carriers of deafness-susceptible genes were identified, with an overall carriage rate of 5.38%. Stratified analysis by altitude showed that: in medium-altitude areas (with a sample size of 2,319 cases), 132 carriers were detected, corresponding to a carriage detection rate of 5.69%; in high-altitude areas (with a sample size of 742 cases), 33 carriers were detected, with a carriage detection rate of 4.45%. Statistical analysis indicated that the carriage detection rate of deafness-susceptible genes among newborns in high-altitude areas was significantly lower than that in medium-altitude areas.

Genotyping data showed: GJB2 gene: A total of 73 mutations (2.38%) were detected, with the 235 del C mutation site being the most common, accounting for 67.12% of the total mutations in this gene. Among them, 58 cases were detected in medium-altitude areas and 15 cases in high-altitude areas, accounting for 20.55% (15/73) of the total GJB2 gene mutations. SLC26A4 gene: A total of 67 mutations (2.19%) were detected, with the IVS7−2 mutation site having the highest frequency, accounting for 47.76% of the total mutations in this gene. Among them, 59 cases were detected in medium-altitude areas and 8 cases in high-altitude areas, accounting for 11.94% (8/67) of the total SLC26A4 gene mutations. Mitochondrial 12S rRNA gene: A total of 33 mutations (1.08%) were detected, with the 1555 A>G mutation site being the main one, accounting for 96.97% of the total mutations in this gene. Among them, 25 cases were detected in medium-altitude areas and 8 cases in high-altitude areas, accounting for 24.24% (8/33) of the total mitochondrial 12S rRNA gene mutations. GJB3 gene: Only 1 rare mutation (0.03%) was detected at the 538 C>T site, and no such mutation was found in high-altitude areas. Analysis of the ethnic distribution of the 3,066 neonates showed: Han ethnicity: A total of 1,983 cases, with 115 cases detected (5.80%), accounting for 69.70% of all carriers. Hui ethnicity: A total of 476 cases, with 24 cases detected (5.04%), accounting for 14.55% of all carriers. Tibetan

ethnicity: A total of 535 cases, with 24 cases detected (4.49%), accounting for 14.55% of all carriers. Mongolian ethnicity: A total of 24 cases, with 2 cases detected (8.33%), accounting for 1.21% of all carriers.

## Conclusion

1. Overall level: The overall carriage rate of deafness susceptibility genes in neonates from Xining, Qinghai is significantly lower than the national average for newborns (6.67%). Within the medium and high altitude regions of Qinghai Province, compared with the medium-altitude areas (altitude > 1,500 meters and ≤ 2,500 meters), the carriage rate of deafness susceptibility genes in neonates from high-altitude areas (altitude > 2,500 meters and ≤ 4,500 meters) shows a downward trend, which suggests that within the medium and high altitude range of the province, the detection rate of deafness susceptibility genes decreases with the increase of altitude. 2. Gene level: Among GJB2 gene mutations, the 235 delC gene mutation remains the most dominant type, and the detection rates of both GJB2 and GJB3 genes are lower than the national level. Notably, the detection rates of mitochondrial 12S rRNA gene mutations and SLC26A4 gene mutations in neonates from this region are both higher than the national average, with the detection rate of mitochondrial 12S rRNA gene mutations being more significantly higher. Since deafness caused by these two types of gene mutations can be effectively reduced in terms of onset risk through health education, conducting genetic testing targeting these specific genes has important clinical significance and public health value. 3. Ethnic background level: The detection rate of deafness susceptibility genes in Tibetan neonates is low, especially for the IVS7−2 A > G mutation site, and this phenomenon is more prominent in high-altitude areas. On the contrary, the detection rate of deafness susceptibility genes in Mongolian and Hui neonates in high-altitude areas shows an increasing trend.

## 1. Introduction

Hereditary deafness is a common birth defect, with a global incidence rate of 2.00‰ to 3.00‰ [1]. In China, the average incidence rate of neonatal deafness ranges from 1.00‰ to 3.47‰ [2,3]. This disease seriously impairs children's hearing and speech development and exhibits a certain degree of concealment [3,4]. Simple neonatal hearing screening is difficult to detect carriers of deafness susceptibility genes, which is not conducive to the early prediction and intervention of children with delayed-onset deafness or drug-induced deafness. In the early stage of hearing loss, parents may mistakenly regard their children's manifestations as slow reaction or overlook the symptoms of hearing loss; by the time the problem is identified, the optimal period for speech development has often been missed.

Screening for hereditary deafness susceptibility genes is a key measure to prevent deafness-related birth defects and reduce the incidence of hearing and speech disabilities. Among the Chinese population, there is a clear ranking of detection rates for common deafness susceptibility genes: in descending order of detection rate, they are GJB2 gene mutations, SLC26A4 gene mutations, mitochondrial 12SrRNA gene mutations, and GJB3 gene mutations. From a clinical perspective, deafness caused by these four gene mutations accounts for more than 80% of all hereditary deafness cases, among which deafness induced solely by GJB2 gene mutations accounts for approximately 55% [5].

Through precise classification of deafness based on distinct genetic profiles, it is feasible to predict the incidence probability, manifestations, and potential precipitating factors of neonatal deafness from a genetic and susceptibility perspective. Building on this foundation, targeted interventions can be implemented at the etiological level, and more importantly, preemptive interventions through health education can play a pivotal role. For example, for carriers of mitochondrial 12SrRNA gene mutations, avoiding ototoxic drugs under the guidance of health education can effectively prevent drug-induced deafness; for individuals with SLC26A4 gene mutations, enhanced health management such as trauma prevention through health education can reduce the risk of delayed-onset deafness to a certain extent. Such preventive health

education combined with etiological interventions collectively forms an integrated comprehensive intervention system encompassing prevention, control, and treatment. Furthermore, China has a vast territory, and significant differences in the incidence rate of hereditary deafness and the types of detected genes may exist across different regions-especially between the central and eastern plains and the western plateaus-and at different altitudes. The western plateaus of China are centered on the Qinghai-Tibet Plateau, with an average altitude exceeding 4000 meters, while the central and eastern plains are mostly below 200 meters above sea level. This geographical variation may influence the genetic characteristics of the disease, further highlighting the necessity of identifying high-risk populations through deafness susceptibility gene screening and precisely matching health education and intervention measures to address regional disparities.

Previous relevant studies have mostly been based on data from the central and eastern plains of China-a region with a dense population and abundant accumulated medical data-whereas research data on neonatal deafness genes in the western plateau areas (mainly the Qinghai-Tibet Plateau and its surrounding regions) is relatively scarce. This study was conducted in Xining, the capital city of Qinghai Province in China. Located in a mid-to-high altitude hypoxic region (at an altitude of 2261–4394 meters), Xining is situated in the northeastern part of the Qinghai-Tibet Plateau and serves as a typical representative of a high-altitude hypoxic environment. The study aims to clarify the characteristics of neonatal deafness genes in a high-altitude hypoxic environment and provide basic data to supplement research in this field in plateau areas.

This study conducted multi-angle and multi-level analysis and interpretation of the deafness susceptibility gene screening results of 3066 newborns at the Affiliated Hospital of Qinghai University. Qinghai Province is located in northwestern China and is an important part of the Qinghai-Tibet Plateau, with significant altitude differences within the province (ranging from approximately 2000 meters in the eastern river valleys to over 4500 meters in the southern plateaus). The study reveals the common deafness gene mutation sites and the carrier rate of deafness gene mutations in newborns at different altitudes in the Xining area of Qinghai Province from various perspectives. It provides a basis and direction for the prevention and control of hereditary deafness in newborns in Qinghai Province, and standardizes the workflow of deafness gene screening and post-screening, so as to more effectively serve the early diagnosis, treatment, and prevention of neonatal deafness.

## 2. Data and methods

### 2.1. Data

This study employed the microarray chip method to detect deafness-susceptibility genes in dried blood spots. The study has been approved by the Scientific Research Ethics Committee of The Affiliated Hospital of Qinghai University (approval number: P-SL-2023181) and has obtained the approval for the collection of Chinese human genetic resources (approval number: 2022CJ1661). After fully obtaining the written informed consent of the newborns' parents, the research team conducted systematic and continuous blood sampling from March 2023 to March 2025 on newborns who were born in the Department of Neonatology of The Affiliated Hospital of Qinghai University and within 48–72 hours after birth. A total of 3,615 heel blood specimens were collected. After quality inspection, 549 unqualified specimens were excluded, and 3,066 qualified specimens were finally included. Among these, 1,585 specimens were from male infants and 1,481 from female infants. The researchers made dried blood spots from the collected heel blood, and then completed the detection of deafness-susceptibility genes in the dried blood spots using the microarray chip method.

### 2.2. Classification criteria

**2.2.1. Classified by different altitudes.** According to Plateau Medicine (Peking University Medical Press) and the expert consensus of the 6th International Conference on High Altitude Medicine and Hypoxic Physiology: low-altitude areas refer to regions with an altitude of 500–1500 meters; medium-altitude areas are regions with an altitude of 1500–

2500 meters; high-altitude areas are regions with an altitude of 2500–4500 meters; and extremely high-altitude areas are regions with an altitude of 4500–5500 meters [6,7].

**2.2.2. Classification by ethnic groups involved in the existing specimens.** Based on the ethnic backgrounds of the collected newborns, the 3,066 newborns were divided into Han, Tibetan, Hui, Mongolian, Tu, Salar, Buyi, Miao, Dongxiang, Kazak, Manchu, Achang, Uyghur, and Zhuang nationalities.

## 2.3. Sample collection and preservation

Within 48 hours postpartum, 3 drops of heel blood are collected from newborns in accordance with the standard protocol [8]. Each blood spot must be greater than 8 mm in diameter and treated with ethylenediaminetetraacetic acid (EDTA) as an anticoagulant. The collection cards are then air-dried to create dried blood spot cards, which are subsequently transported to the National Gene Detection Technology Application Demonstration Center at the South Campus of Qinghai University Affiliated Hospital. These samples are stored at 4°C in a refrigerator, awaiting analysis.

## 2.4. Instruments and reagents

Gene amplifier (Bioer TC-96/G/H (b)), chip washing and drying instrument (CapitalBio SlideWasherTM 24), microarray chip scanner (CapitalBio LuxScanTM 10K-B), centrifuge (Eppendorf 5424), constant temperature shaking metal bath (Bioer MB-102).

Fifteen-item hereditary deafness-related gene detection kit (microarray chip method, article number: 300068−01) was purchased from Beijing CapitalBio Technology Co., Ltd.

## 2.5. Newborn deafness genetic screening

A 3 mm-diameter blood sample was collected using an automated punching instrument, followed by DNA extraction, sequence amplification, and library preparation in sequence, and then detection was completed using a genetic sequencer. During the detection process, a deafness gene detection kit was used to accurately screen 15 loci of 4 common deafness susceptibility genes, including GJB2 gene (35 del G, 176_191 del 16, 235 del C, 299_300 del AT), SLC26A4 gene (2168 A>G, IVS7−2 A>G, 1174 A>T, 1226 G>A, 1229 C>T, IVS15+5 G>A, 1975 G>C, 2027 T>A), mitochondrial 12SrRNA gene (1494 C>T, 1555 A>G) and GJB3 gene (c.538C>T). To ensure the accuracy and reliability of the detection results, all initially screened positive samples must be further confirmed through nucleic acid re-examination, so as to minimize the interference of false positive results on clinical judgment and provide a solid basis for subsequent interventions.

## 2.6. Statistical methods

Microsoft Excel was used for data measurement and preliminary statistical analysis. Categorical data were expressed as n (%), and the Adjusted Wald method was employed to calculate the corresponding 95% confidence intervals (95% CIs). Statistical analyses and data visualization were performed using SPSS version 31.0 and GraphPad Prism version 10. A P-value of < 0.05 was considered statistically significant.

## 3. Results

### 3.1. The overall detection of deafness susceptibility genes

Among 3,066 qualified newborn blood samples, a total of 165 cases with deafness gene mutations were screened out, with a mutation detection rate of 5.38% (165/3,066). Among them, the heterozygous mutation rate was 4.83% (148/3,066), accounting for 89.70% of the positive detection results; the homozygous mutation rate was 0.55% (17/3,066) (Fig 1). It can be seen that the detection of deafness susceptibility gene carriers is mainly heterozygous.

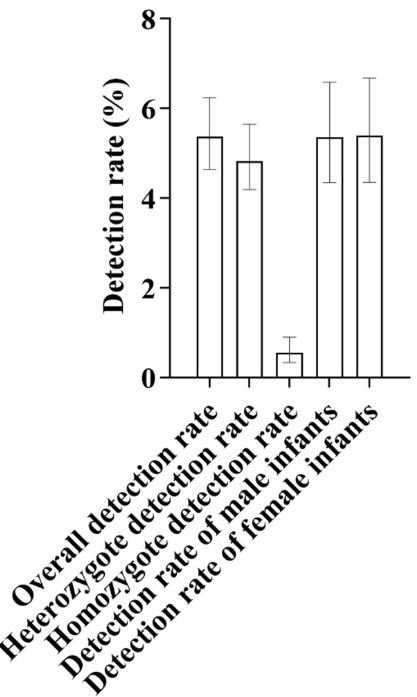

**Fig 1. As shown in Figure, the overall detection rate of deafness-susceptible genes in this study was 5.38%.** Regarding genotype distribution, heterozygotes accounted for 4.83%, while homozygotes accounted for 0.55%, indicating that heterozygotes were the predominant type detected. In terms of gender, the carrier rate of deafness-susceptible genes was 5.36% in male infants and 5.40% in female infants. The rates were very similar, with only a 0.04% difference between males and females, both aligning closely with the overall detection rate (5.38%). The bars in the figure indicate the 95% confidence intervals (95% CIs) for each group's results.

In terms of gender distribution, there were 1,585 male infants in total, with a mutation detection rate of 5.36% (85/1,585); there were 1,481 female infants in total, with a mutation detection rate of 5.40% (80/1,481) (Fig 1). The mutation detection rates between male and female infants were basically the same.

Specifically, regarding the mutation detection of each gene: the detection rate of GJB2 gene mutation was 2.38%, that of SLC26A4 gene mutation was 2.19%, that of mitochondrial 12S rRNA gene mutation was 1.08%, and that of GJB3 gene mutation was 0.03% (Fig 2 and Table 1).

Fig 3 shows the typical sequencing diagrams of four genes, namely GJB2, SLC26A4, mitochondrial 12SrRNA and GJB3.

### 3.2. GJB2 gene mutation

In neonates in Qinghai region, the detection of GJB2 gene mutations is as follows: a total of 73 cases were detected, with a detection rate of 2.38% (73/3066). The incidence of this gene mutation ranks first among all types of gene mutations in local neonates.

Among the 73 detected cases of GJB2 gene mutations, there are 36 male infants and 37 female infants. In terms of ethnic distribution, 50 cases are Han (detection rate 2.52%, 50/1983), 12 cases are Tibetan (detection rate 2.24%, 12/535), 9 cases are Hui (detection rate 1.89%, 9/476), and 2 cases are Mongolian (detection rate 8.33%, 2/24) (Fig 4). From the perspective of altitude distribution, 58 cases were detected in medium-altitude areas (detection rate 2.50%, 58/2319), and 15 cases were detected in high-altitude areas (detection rate 2.02%, 15/742) (Fig 5), accounting for 20.55% (15/73) of the total detected GJB2 gene mutations.

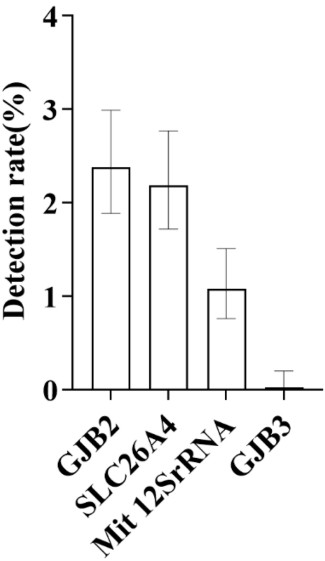

**Fig 2. As shown in Figure, the distribution of detection rates for the four major deafness-related genes-GJB2, SLC26A4, mitochondrial 12S rRNA, and GJB3-is clearly illustrated.** Among these, GJB2 exhibited the highest detection rate at 2.38%, while GJB3 showed the lowest at 0.03%. The bars in the figure indicate the 95% confidence intervals (95% CIs) for each group's results.

**Table 1. Shows the overall detection rates of a total of 15 loci in the four genes screened in this study.**

| Gene | Detection Locus | Locus Mutation detection rate(%) | Gene Mutation detection rate(%) |
|---|---|---|---|
| GJB2 | 35 delG | 0.13% | 2.38% |
| | 176_191 del16 | 0.13% | |
| | 235 delC | 1.60% | |
| | 299_300 delAT | 0.52% | |
| GJB3 | 538 C>T | 0.03% | 0.03% |
| Mit12SrRNA | 1494 C>T | 0.03% | 1.08% |
| | 1555 A>G | 1.04% | |
| SLC26A4 | 1174 A>T | 0.16% | 2.19% |
| | 1226 G>A | 0.26% | |
| | 1229 C>T | 0.10% | |
| | 1975 G>C | 0.16% | |
| | 2027 T>A | 0.16% | |
| | 2168 A>G | 0.23% | |
| | IVS7−2 A>G | 1.04% | |
| | IVS15+5 G>A | 0.10% | |

Analysis of mutation sites and types shows that a total of 4 GJB2 gene mutation sites were detected in this study, including 72 cases of heterozygous mutations and only 1 case of homozygous mutation. Among all mutation sites, the mutation rate of c.235delC is the highest, reaching 1.60% (49/3066), accounting for 67.12% of GJB2 gene mutations. Its distribution is as follows: 26 male infants and 23 female infants; 37 Han, 7 Tibetan, 4 Hui, and 1 Mongolian; 38 cases in medium-altitude areas and 11 cases in high-altitude areas. The next is the c.299_300delAT site, with a detection rate of

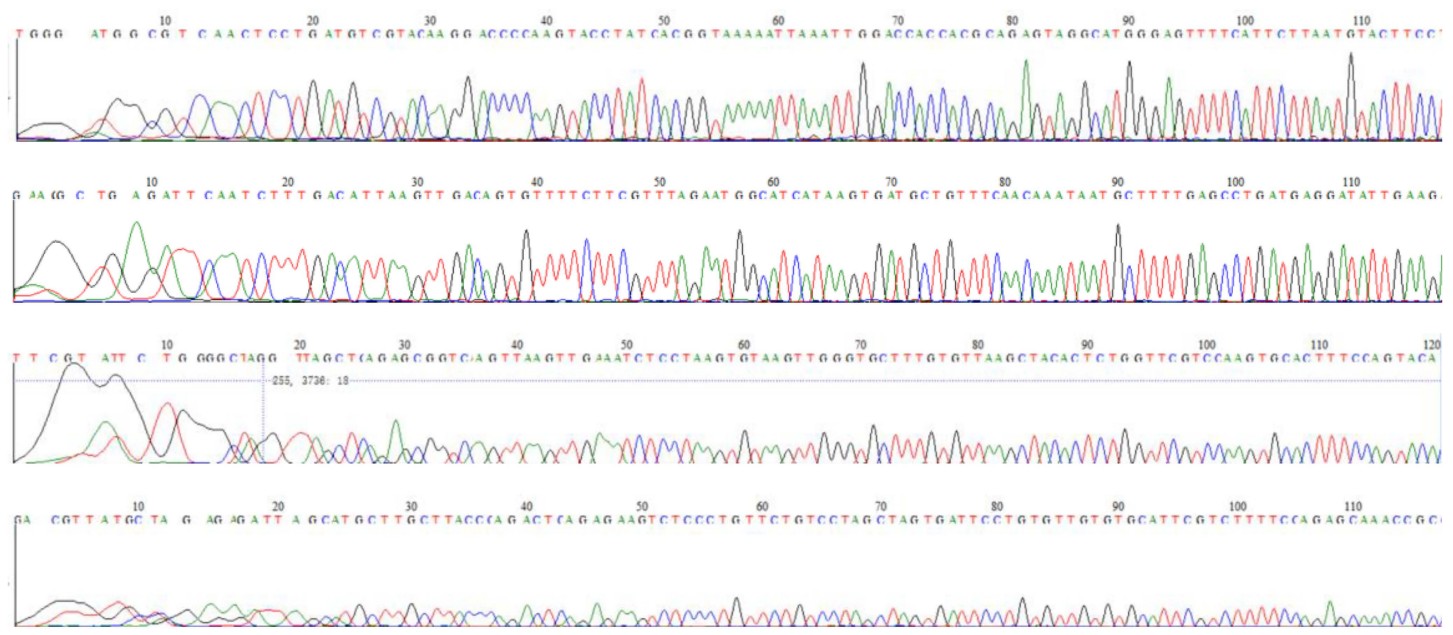

**Fig 3. The four figures above show the base sequence maps and locus mutations of the detected deafness-susceptible genes after rechecking.**

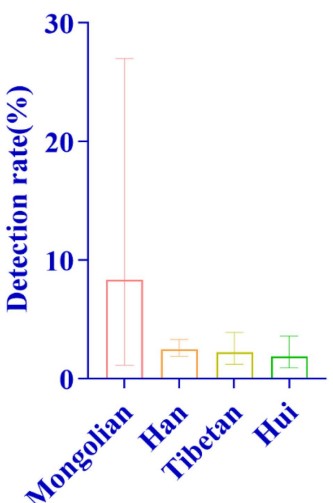

**Fig 4. Ethnic Distribution of GJB2 Gene Mutation Carriers: As illustrated in the figure, among different ethnic groups, the detection rate of GJB2 gene mutations was highest in Mongolians, followed by Han, Tibetans, and Hui populations.** The bars in the figure indicate the 95% confidence intervals (95% CIs) for each group's results.

0.52% (16/3066). The distribution is: 6 male infants and 10 female infants; 7 Han, 5 Tibetan, 3 Hui, and 1 Mongolian; 13 cases in medium-altitude areas and 3 cases in high-altitude areas. There are 4 cases of c.35delG heterozygous variation (detection rate 0.13%, 4/3066), including 2 male infants and 2 female infants; 3 Han and 1 Hui (no Tibetan cases detected); 3 cases in medium-altitude areas and 1 case in high-altitude areas. There are 4 cases of c.176_191del16

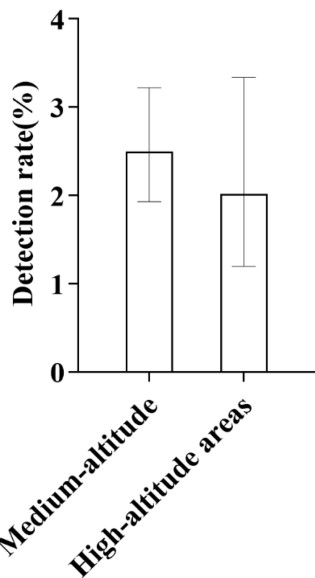

**Fig 5. Altitude Distribution Characteristics of GJB2 Gene Mutations: As shown in the figure, GJB2 gene mutations were detected in both medium-altitude and high-altitude regions, with a relatively lower mutation rate observed in high-altitude areas.** The bars in the figure represent the 95% confidence intervals (95% CIs) for each group, and this convention applies throughout unless otherwise stated.

heterozygous variation (all from medium-altitude areas), with a detection rate of 0.13% (4/3066), including 2 male infants and 2 female infants; 3 Han and 1 Hui (no Tibetan cases detected) (Fig 6).

### 3.3. SLC26A4 gene mutation

The detection results of SLC26A4 gene mutations in the newborn population in Qinghai are as follows: A total of 67 cases were detected, with a detection rate of 2.19% (67/3066). Among them, there were 65 heterozygotes and 2 homozygotes; 35 were male infants and 32 were female infants.

In terms of ethnic distribution, 49 cases were detected in the Han nationality, with a detection rate of 2.47% (49/1983); 5 cases were detected in the Tibetan nationality, with a detection rate of 0.93% (5/535); 13 cases were detected in the Hui nationality, with a detection rate of 2.73% (13/476) (Fig 7). In terms of distribution by altitude regions, 59 cases were detected in medium-altitude areas, with a detection rate of 2.54% (59/2319); 8 cases were detected in high-altitude areas, with a detection rate of 1.08% (8/742) (Fig 8). The detection rate of this gene mutation ranks second among all types of gene mutations in local newborns.

**2.3.1. Specifically for mutation sites, sorted by mutation rate from high to low as follows (Fig 9).** c.IVS7−2 A>G site: A total of 32 cases were detected, with a detection rate of 1.04%, accounting for 47.76% of the total number of people detected with SLC26A4 gene mutations. Among them, there were 20 male infants (detection rate 1.26%) and 12 female infants (detection rate 0.81%), and the detection rate in male infants was higher than that in female infants; 22 cases were in the Han nationality (detection rate 1.11%), 2 cases were in the Tibetan nationality (detection rate 0.37%), and 8 cases were in the Hui nationality (detection rate 1.68%); 26 cases were detected in medium-altitude areas (detection rate 1.12%, 26/2319), and 6 cases were detected in high-altitude areas (detection rate 0.81%, 6/742). c.1226 G>A site: 8 cases were detected, all from medium-altitude areas, with a detection rate of 0.26%. Among them, there were 4 male infants and 4 female infants; 5 cases were in the Han nationality (detection rate 0.25%) and 3 cases were in the Hui nationality (detection rate 0.63%); no mutations at this site were detected in the Tibetan population or high-altitude

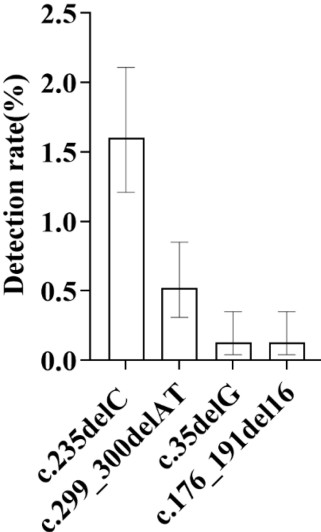

**Fig 6. Distribution of Mutation Rates at Four GJB2 Gene Loci: As shown in the figure, among all mutation loci of the GJB2 gene, the c.235delC locus exhibited the highest mutation rate at 1.60%, followed by the c.299_300delAT locus with a detection rate of 0.52%.** The c.35delG heterozygous variant and the c.176_191del16 heterozygous variant showed the same detection rate, both at 0.13%.

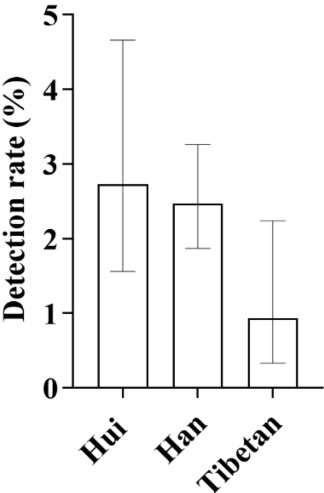

**Fig 7. Ethnic Distribution Characteristics of SLC26A4 Gene Mutation Detection Rates: As shown in the figure, among different ethnic groups, the detection rate of SLC26A4 gene mutations was highest in the Hui population, surpassing that of the Han group for the first time.** In contrast, the detection rate in the Tibetan population was significantly lower and remained at a relatively low level.

areas. c.2168 A>G site: 7 cases were detected, with a detection rate of 0.23%. Among them, there were 3 male infants and 4 female infants; 4 cases were in the Han nationality (detection rate 0.20%), 2 cases were in the Tibetan nationality (detection rate 0.37%), and 1 case was in the Hui nationality (detection rate 0.21%); 5 cases were detected in medium-altitude areas (detection rate 0.22%), and 2 cases were detected in high-altitude areas (detection rate 0.27%). c.1174 A>T, 1975 G>C and 2027 T>A sites: 5 cases were detected at each of these three sites, with a detection rate of 0.16% for each. Among them, there were 6 male infants and 9 female infants; 14 cases were in the Han nationality, 1 case was

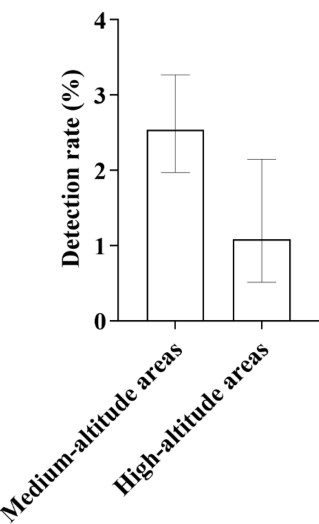

**Fig 8. Distribution of SLC26A4 Gene Detection Rates Across Different Altitude Regions: As shown in the figure, the detection rate of the SLC26A4 gene exhibited significant altitude-related variation.** Specifically, the detection rate in high-altitude areas was lower than that in medium-altitude areas. This distribution pattern was consistent with the trend observed for GJB2 gene detection rates across different altitude regions. For medium-altitude areas, the 95% confidence interval (95% CI) ranged from 0.0197 to 0.0327, while for high-altitude areas, it ranged from 0.0051 to 0.0215.

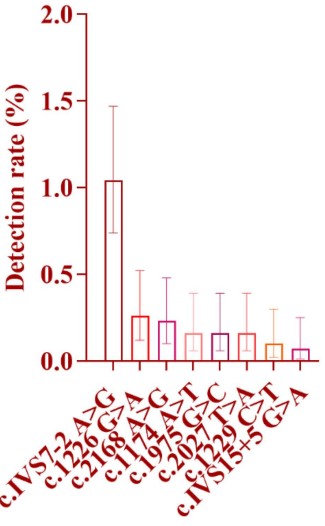

**Fig 9. Distribution of Detection Rates for Eight Mutation Sites in the SLC26A4 Gene: As shown in the figure, among the eight mutation sites of the SLC26A4 gene, the c.IVS7-2A>G site exhibited the highest mutation frequency, with a detection rate of 1.04%, followed by the c.1226G>A site, with a detection rate of 0.23%.** The IVS15+5G>A site showed the lowest detection rate, at only 0.07%.

in the Hui nationality, and no cases were detected in the Tibetan nationality; 15 cases were detected in medium-altitude areas, and no mutations at these three sites were detected in high-altitude areas. c.1229 C>T and IVS15+5 G>A sites: These are the sites with the lowest detection rates among SLC26A4 gene mutations, with 3 cases (detection rate 0.10%) and 2 cases (detection rate 0.07%) detected respectively. Mutations at these two sites were both detected in medium-altitude areas, including 2 male infants and 3 female infants; 4 cases were in the Han nationality and 1 case was in the Tibetan nationality, and no mutations at these two sites were detected in high-altitude areas either.

## 3.4. Mitochondrial 12SrRNA gene mutation

Detection results of mitochondrial 12S rRNA gene mutations in Qinghai newborns showed that a total of 33 mutation cases were detected, with an overall detection rate of 1.08% (33/3066). Among them, there were 19 cases of hetero-plasmic mutations, with a detection rate of 0.62%; and 14 cases of homoplasmic mutations, with a detection rate of 0.46% (Fig 10).

In terms of demographic characteristics, 17 male infants and 16 female infants were detected, showing a relatively balanced gender distribution. From the ethnic perspective, 19 cases were found in Han nationality, with a detection rate of 0.96%; 8 cases in Tibetan nationality, with a detection rate of 1.50%; and 5 cases in Hui nationality, with a detection rate of 1.05%. It was the first time that the detection rate of deafness-susceptible genes in Tibetan population exceeded that in Han population (Fig 11). According to the altitude division, 25 cases were detected in medium-altitude areas and 8 cases in high-altitude areas, with the detection rate of both being 1.08%, showing no statistical difference (Fig 12). The detection rate of this gene mutation ranked the third among all types of gene mutations in local newborns.

In terms of mutation sites, the mutation rate of c.1555 A>G site was the highest, with a detection rate of 1.04%, accounting for 96.97% of the total cases of mitochondrial 12S rRNA gene mutations. Its distribution among different gen-ders and nationalities was as follows: 16 cases in male infants and 16 cases in female infants; 19 cases in Han national-ity, 8 cases in Tibetan nationality and 5 cases in Hui nationality. The second was the mutation at c.1494 C>T site, with a detection rate of 0.03% (Fig 13), and no such mutation was found in high-altitude areas.

## 3.5. GJB3 gene mutation

Only one case of GJB3 gene mutation was detected in the middle-altitude area, which was a c.538 C>T site mutation, with a detection rate of 0.03% (1/3066).

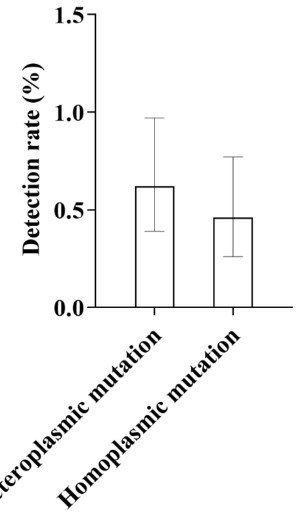

**Fig 10. Distribution of Detection Rates for Homoplasmic and Heteroplasmic Mutations in the Mitochondrial 12S rRNA Gene: As shown in the figure, among the mutation types of the mitochondrial 12S rRNA gene, the detection rate of heteroplasmic mutations was relatively higher, at 0.62%, while that of homoplasmic mutations was 0.46%.** A noticeable difference in mutation frequency was observed between the two types.

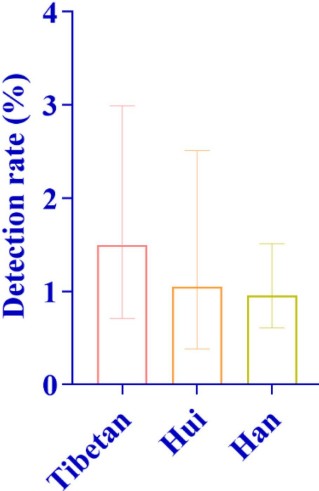

**Fig 11. Distribution of Detection Rates of Mitochondrial 12S rRNA Gene Mutations Among Different Ethnic Groups: As shown in the figure, the detection rates of mitochondrial 12S rRNA gene mutations varied among different ethnic groups.** The highest detection rate was observed in the Tibetan population, followed by the Hui population, while the Han population exhibited the lowest detection rate.

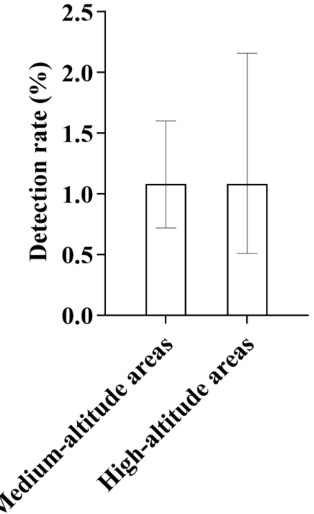

**Fig 12. Distribution of Detection Rates of Mitochondrial 12S rRNA Gene Mutations in High- and Medium-Altitude Areas: As shown in the figure, the detection rates of mitochondrial 12S rRNA gene mutations were identical in both high-altitude and medium-altitude regions, each being 1.08%.** These results suggest that there was no statistically significant difference in the detection rates of this gene mutation between the two altitude regions.

### 3.6. Analysis of detection results of deafness-susceptible genes and their loci in newborns with different ethnic backgrounds

This study summarizes and analyzes the screening results of deafness-susceptible genes in newborns with different ethnic backgrounds, covering 14 ethnic groups including the Han, Tibetan, Hui, Mongolian, and Tu nationalities. The specific detection results are as follows:

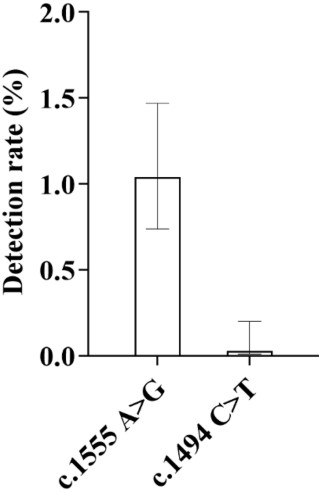

**Fig 13. Distribution of Detection Rates of Major Mutation Sites in the Mitochondrial 12S rRNA Gene: As shown in the figure, among the two detected mutation sites of this gene, the c.1555A>G site exhibited a clearly dominant mutation frequency, with a detection rate of 1.04%.** In contrast, the c.1494C>T site showed a markedly lower detection rate of only 0.03%, indicating a significant difference in mutation frequency between the two sites.

### 3.6.1. General screening overview.

A total of 3,066 neonates from different ethnic groups were enrolled in this study, with 165 cases of deafness-susceptible gene mutations detected. The overall detection results revealed significant differences among different ethnic groups: among them, the number of carriers with deafness-susceptible gene mutations detected in the Han nationality was the largest, accounting for 69.70% of the total detected mutations; the proportions of detected mutations in the Tibetan and Hui nationalities were the same, both at 14.55%; the detection proportion in the Mongolian nationality was relatively low (1.21%), yet its detection rate was the highest, reaching 8.33%, followed by the Han nationality (5.80%); no related mutations were detected in the remaining 10 ethnic groups (Fig 14A and 14B). In addition, in different altitude areas, there were also differences in the detection rates of deafness gene mutations among neonate populations with different ethnic backgrounds (Fig 15).

### 3.6.2. Specific detection results by ethnic group.

**3.6.2.1.Han Nationality**

**3.6.2.1.1. General situation**

A total of 1,983 cases were included, with 115 cases detected, resulting in a detection rate of 5.80%, accounting for 69.70% of the total detected cases (115/165).

**3.6.2.1.2. Gender distribution**

56 male infants and 59 female infants were detected, with no significant difference in detection rates between genders.

**3.6.2.1.3. Altitude difference**

95 cases were detected in the medium-altitude area, with a detection rate of 5.95% (95/1,597); 20 cases were detected in the high-altitude area, with a detection rate of 5.18% (20/386). The detection rate in the medium-altitude area was slightly higher than that in the high-altitude area.

**3.6.2.1.4. Gene mutation loci (Fig 16)**

GJB2 gene: 37 cases were detected at the c.235 del C locus, with a detection rate of 1.87%, which was the highest in the Han nationality; 7 cases were detected at the c.299_300 del AT locus, with a detection rate of 0.35%; 3 cases each were detected at the c.35 del G locus and c.176_191 del 16 locus, with a detection rate of 0.15% for both.

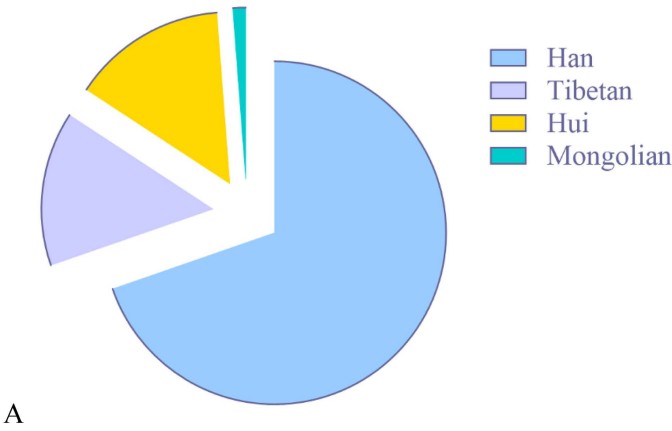

A

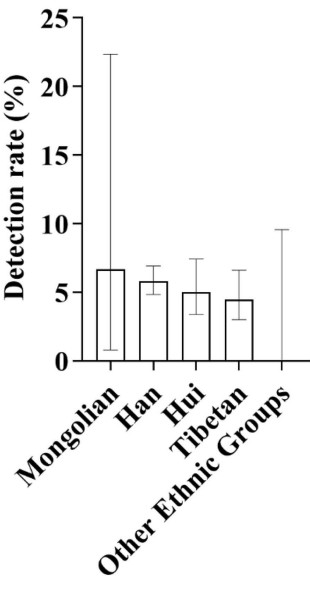

B

**Fig 14. Distribution and detection rates of deafness-susceptible genes among different ethnic groups. A:** Distribution of the proportions of detected cases carrying deafness-susceptible genes among different ethnic groups. Among these groups, the Han population accounted for the largest proportion, reaching 69.70%, followed by the Tibetan and Hui populations, each representing 14.55%. The Mongolian population accounted for 1.21%. Additionally, 10 ethnic groups-including the Tu, Salar, Manchu, Kazakh, Buyi, Achang, Dongxiang, Uyghur, Miao, and Zhuang-had no detected carriers of deafness-susceptible genes and are therefore not shown in the figure. **B:** Comparison of the overall detection rates of deafness-susceptible genes among different ethnic groups. The results show that the Mongolian population exhibited the highest detection rate, at 6.67%, followed by the Han (5.80%), Hui (5.04%), and Tibetan (4.49%) populations. No deafness-susceptible genes were detected in the remaining ten ethnic groups.

SLC26A4 gene: 22 cases were detected at the c.IVS7−2 A>G locus, with a detection rate of 1.11%, which was the second highest after the c.235 del C locus; 5 cases each were detected at the c.1174 A>T, 1226 G>A, and 1975 G>C loci, with a detection rate of 0.25% for each; 4 cases each were detected at the c.2027 T>A and 2168 A>G loci, with a detection rate of 0.20%; 2 cases each were detected at the c.1229 C>T and IVS15+5 G>A loci, with a detection rate of 0.10%.

Mitochondrial 12S rRNA gene: 19 cases were detected at the c.1555 A>G locus, with a detection rate of 0.96%; 1 case was detected at the c.1494 C>T locus, with a detection rate of 0.05%.

GJB3 gene: 1 case was detected at the c.538 C>T locus, with a detection rate of 0.05%.

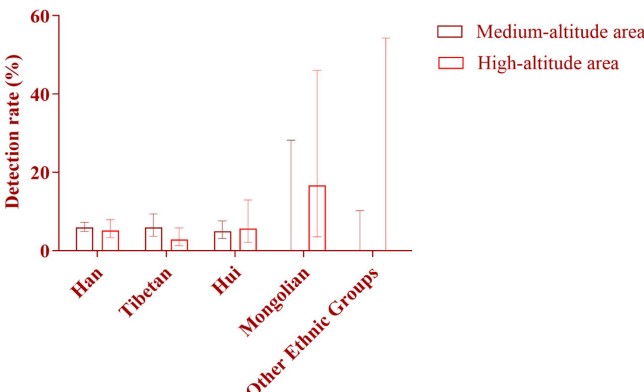

**Fig 15. Comparison of Detection Rates of Deafness Genes Between Medium- and High-Altitude Areas Among Different Ethnic Groups: As shown in the figure, among Han and Tibetan newborns, the detection rates of deafness genes in high-altitude areas were lower than those in medium-altitude areas, with the rate among Tibetan newborns being particularly low.** In contrast, among Mongolian and Hui newborns, the detection rates in high-altitude areas were higher than those in medium-altitude areas, with a more pronounced difference observed in the Mongolian population. No deafness-susceptible genes were detected in the remaining ten ethnic groups.

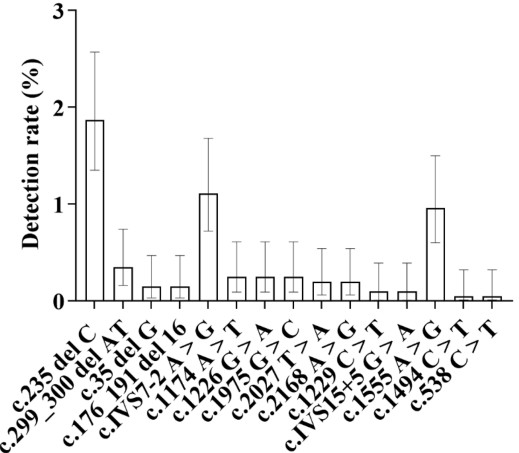

**Fig 16. Distribution of Detection Rates of Various Gene Loci Among Han Newborns: As shown in the figure, the c.235delC locus of the GJB2 gene exhibited the highest detection rate among the Han population, reaching 1.87%.** This was followed by the **c.**IVS7-2A>G locus of the SLC26A4 gene, with a detection rate of 1.11%, and the **c.**1555A>G locus of the mitochondrial 12S rRNA gene, with a detection rate of 0.96%.

### 3.6.3. Tibetan nationality.

### 3.6.3.1. General situation

A total of 535 cases were included, with 24 cases detected, resulting in a detection rate of 4.49%, accounting for 14.55% of the total detected cases (24/165).

### 3.6.3.2. Gender distribution

13 male infants and 11 female infants were detected, with a small difference in detection rates between genders.

### 3.6.3.3. Altitude difference

17 cases were detected in the medium-altitude area, with a detection rate of 5.94%; 7 cases were detected in the high-altitude area, with a detection rate of 2.81%. The detection rate in the medium-altitude area was significantly higher than that in the high-altitude area.

#### 3.6.3.4.Gene mutation loci (Fig 17)

GJB2 gene: 7 cases were detected at the c.235 del C locus, with a detection rate of 1.31%, which was the second highest after the mitochondrial 12S rRNA gene c.1555 A>G locus; 5 cases were detected at the c.299_300 del AT locus, with a detection rate of 0.93%; no cases were detected at the c.35 del G locus or c.176_191 del 16 locus. SLC26A4 gene: 2 cases each were detected at the c.IVS7−2 A>G and 2168 A>G loci, with a detection rate of 0.37% for each; 1 case was detected at the c.1229 C>T locus, with a detection rate of 0.19%; no cases were detected at the c.1174 A>T, 1226 G>A, 2027 T>A, 1975 G>C, or IVS15+5 G>A loci. Mitochondrial 12S rRNA gene: 8 cases were detected at the c.1555 A>G locus, with a detection rate of 1.50%, which was the highest in the Tibetan nationality; no cases were detected at the c.1494 C>T locus. GJB3 gene: No cases were detected at the c.538 C>T locus.

#### 3.6.4. Hui nationality.

#### 3.6.4.1. General situation

A total of 476 cases were included, with 24 cases detected, resulting in a detection rate of 5.04%, accounting for 14.55% of the total detected cases (24/165).

#### 3.6.4.2.Gender distribution

16 male infants were detected, with a detection rate of 6.30% (16/254); 8 female infants were detected, with a detection rate of 3.60% (8/222). The detection rate in male infants was higher than that in female infants (Fig 18).

#### 3.6.4.3.Altitude difference

19 cases were detected in the medium-altitude area, with a detection rate of 4.90% (19/388); 5 cases were detected in the high-altitude area, with a detection rate of 5.68% (5/88). The detection rate in the high-altitude area was slightly higher than that in the medium-altitude area.

#### 2.6.4.4.Gene mutation loci (Fig 19)

GJB2 gene: 4 cases were detected at the c.235 del C locus, with a detection rate of 0.84%; 3 cases were detected at the c.299_300 del AT locus, with a detection rate of 0.63%; 1 case each was detected at the c.35 del G locus and c.176_191 del 16 locus, with a detection rate of 0.21% for each. SLC26A4 gene: 8 cases were detected at the c.IVS7−2 A>G locus, with a detection rate of 1.68%, which was the highest in the Hui nationality; 3 cases were detected at the 1226 G>A locus, with a detection rate of 0.63%; 1 case each was detected at the c.2027 T>A and 2168 A>G loci, with a detection rate of 0.21% for each; no cases were detected at the c.1174 A>T, 1975 G>C, 1229 C>T, or IVS15+5 G>A loci. Mitochondrial 12S rRNA gene: 5 cases were detected at the c.1555 A>G locus, with a detection rate of 1.05%, which was the second highest after the c.IVS7−2 A>G locus; no cases were detected at the c.1494 C>T locus. GJB3 gene: No cases were detected at the c.538 C>T locus.

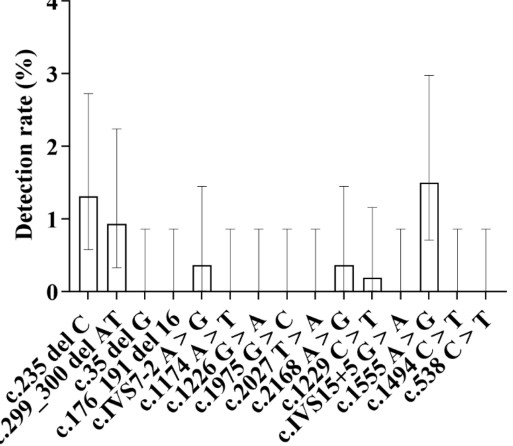

**Fig 17. Distribution of Detection Rates of Various Deafness-Susceptible Gene Mutation Loci in Tibetan Newborns: As shown in the figure, among the Tibetan population, the c.1555A>G locus in the mitochondrial 12S rRNA gene exhibited the highest detection rate, reaching 1.50%.** This was followed by the c.235delC locus in the GJB2 gene, with a detection rate of 1.31%, and the c.299_300delAT locus, with a detection rate of 0.93%.

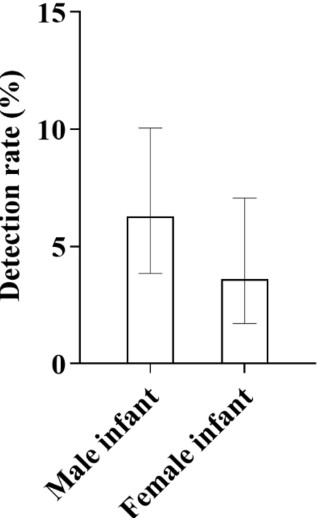

**Fig 18. Comparison of Detection Rates of Deafness Genes Between Male and Female Newborns in the Hui Screening Population: As shown in the figure, the detection rate of deafness genes in male infants (6.30%) was notably higher than that in female infants (3.60%).**

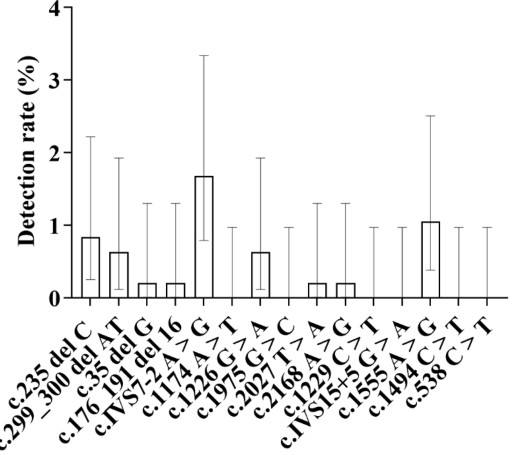

**Fig 19. Detection Rates of Deafness-Susceptible Gene Mutations in Hui Newborns: As shown in the figure, the distribution of detection rates for deafness-susceptible gene mutations among Hui newborns is presented.** The results indicate that the **c.**IVS7-2A>G locus in the SLC26A4 gene exhibited the highest detection rate, reaching 1.68%, followed by the **c.**1555A>G locus in the mitochondrial 12S rRNA gene, with a detection rate of 1.05%. The c.235delC locus in the GJB2 gene ranked third, with a detection rate of 0.84%.

### 3.6.5. Mongolian nationality.

### 3.6.5.1. General situation

A total of 24 Mongolian newborns were included in this study, among which 2 cases tested positive, with an overall detection rate of 8.33%.

### 3.6.5.2. Stratified analysis by gender

Among 15 male infants, 1 case tested positive, with a detection rate of 6.67%; among 9 female infants, 1 case tested positive, with a detection rate of 11.11%, and the detection rate in female infants was significantly higher than that in male infants (Fig 20).

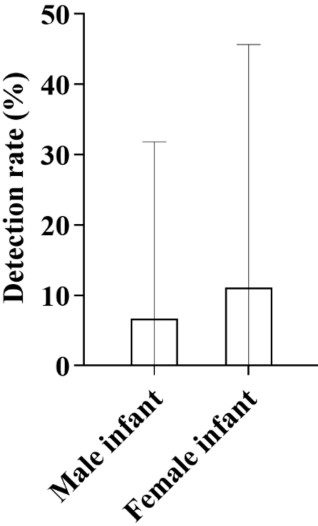

**Fig 20. Comparison of Detection Rates of Deafness-Susceptible Genes Between Male and Female Mongolian Newborns: As shown in the figure, with a sample size of 12 cases in each group, the detection rate of deafness-susceptible genes in female infants was 11.11%, which was higher than that in male infants (6.67%).**

### 3.6.5.3. Stratified analysis by altitude

2 cases were detected in high-altitude areas, with a detection rate of 16.67%; no carriers of deafness gene mutations were detected in medium-altitude areas.

### 3.6.5.4. Detection of gene loci

Only 1 case was detected at each of the c.235 del C locus and c.299_300 del AT locus of the GJB2 gene, with the detection rate of both loci being 4.17%; no positive results were detected at all tested loci of the SLC26A4 gene, mitochondrial 12S rRNA gene, and GJB3 gene (Fig 21).

**3.6.6. Other ethnic groups.** A total of 44 cases were included from the Tu (32 cases), Salar (4 cases), Manchu (4 cases), Kazakh (2 cases), Buyi (1 case), Achang (1 case), Dongxiang (1 case), Uyghur (1 case), Miao (1 case), and Zhuang (1 case) nationalities. No deafness-susceptible gene mutations were detected in these 10 ethnic groups, particularly in the Tu nationality.

**3.6.7. Summary of Ethnic-Specific Key Loci (Detailed information is shown in Table 2).** Among Han newborns, the gene locus with the highest detection rate was the GJB2 gene c.235 del C mutation. Among Tibetan newborns, the gene locus with the highest detection rate was the mitochondrial 12S rRNA gene c.1555 A > G mutation. Among Hui newborns, the gene locus with the highest detection rate was the SLC26A4 gene c.IVS7−2 A > G mutation.

The data in the table shows that the deafness-susceptible gene mutation IVS7−2 A>G, which is highly prevalent in the Han and Hui populations, has a significantly lower detection rate in Tibetan neonates. This phenomenon may, from the perspective of environmental selection, explain why the detection rate of deafness-susceptible genes in neonates in high-altitude areas is lower than that in low-altitude areas.

## 3.7. Analysis of detection characteristics of deafness-susceptible genes in newborns in different altitude regions

Among the 3,066 newborns who participated in the deafness-susceptible gene screening, the distribution and gene detection results according to their long-term residential altitude ranges are analyzed as follows:

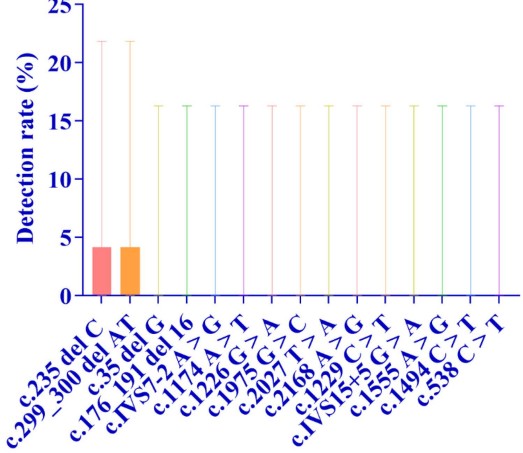

**Fig 21. Distribution of Detection Rates of Deafness-Susceptible Genes in Mongolian Newborns: As shown in the figure, a total of two cases of deafness gene carriers were identified in the Mongolian population, both involving mutations in the GJB2 gene.** Specifically, the mutations were located at the c.235delC and c.299_300delAT loci, each with a detection rate of 4.17% (95% CI: < 0.0001-0.2187). No mutations were detected in other genes or loci.

**3.7.1. Low-altitude areas (500 m < altitude ≤ 1,500 m).** There are 5 newborns in this range, accounting for 0.16% of the total number of screened individuals (Note: Qinghai Province in China belongs to medium and high-altitude areas). Among them, 3 are Han nationality and 2 are Tibetan nationality. The gene detection results show that no deafness mutant genes were detected in low-altitude areas.

**3.7.2. Medium-altitude areas (1,500 m < altitude ≤ 2,500 m). 3.7.2.1. Basic distribution**

There are 2,319 newborns in this interval, accounting for 75.64% of the total number of screened individuals (2,319/3,066). A total of 132 newborns carrying deafness gene mutations were detected in the medium-altitude area, with an overall detection rate of 5.69%, which is higher than that in the high-altitude area (4.45%) (Fig 22). Among them, there are 1,208 male infants and 1,111 female infants.

**3.7.2.2.Ethnic distribution and detection status of newborns in medium-altitude areas**

1,594 Han nationality newborns, 95 cases detected, with a detection rate of 5.96%; 388 Hui nationality newborns, 19 cases detected, with a detection rate of 4.90%; 284 Tibetan nationality newborns, 17 cases detected, with a detection rate of 5.99%; 29 Tu nationality newborns, no detection at all gene loci; 12 Mongolian nationality newborns, 1 case detected, with a detection rate of 8.33%; A total of 12 newborns from the other 7 ethnic groups (4 Salar, 3 Manchu, 1 Buyi, 1 Dongxiang, 1 Achang, 1 Uyghur, and 1 Zhuang), with no deafness mutant genes detected (Fig 23).

**3.7.2.3.Details of detection of various genes and loci in medium-altitude areas (Fig 24)**

**3.7.2.3.1. GJB2 gene**

38 cases detected at the c.235 del C locus, with a detection rate of 1.64%, which is the highest detection rate locus in medium-altitude areas; 13 cases detected at the c.299_300 del AT locus, with a detection rate of 0.56%; 4 cases detected at the c.176_191 del 16 locus, with a detection rate of 0.17%; 3 cases detected at the c.35 del G locus, with a detection rate of 0.13%.

**3.7.2.3.2. SLC26A4 gene**

26 cases detected at the c.IVS7−2 A>G locus, with a detection rate of 1.12%, which is the second highest detection rate locus after the c.235 del C locus; 8 cases detected at the 1226 G>A locus, with a detection rate of 0.34%; 5 cases each detected at the c.1174 A>T, c.1975 G>C, c.2027 T>A, and c.2168 A>G loci, with a detection rate of 0.22% for

**Table 2. Among the 165 neonates confirmed as carriers of deafness mutant genes after rechecking, the number of detected cases at deafness gene mutation loci among neonates of different ethnic groups is as follows.**

| Ethnic Group | Detection Locus | Number of Detected Cases at Mutation Loci(case(s)) |
|---|---|---|
| Han | 35 delG | 3 |
| | 176_191 del16 | 3 |
| | 235 delC | 37 |
| | 299_300 delAT | 7 |
| | 538 C>T | 1 |
| | 1494 C>T | 1 |
| | 1555 A>G | 19 |
| | 1174 A>T | 5 |
| | 1226 G>A | 5 |
| | 1229 C>T | 2 |
| | 1975 G>C | 5 |
| | 2027 T>A | 4 |
| | 2168 A>G | 4 |
| | IVS7−2 A>G | 22 |
| | IVS15+5 G>A | 2 |
| Tibetan | 235 delC | 7 |
| | 299_300 delAT | 5 |
| | 1555 A>G | 8 |
| | 1229 C>T | 1 |
| | 2168 A>G | 2 |
| | IVS7−2 A>G | 2 |
| Hui | 35 delG | 1 |
| | 176_191 del16 | 1 |
| | 235 delC | 4 |
| | 299_300 delAT | 3 |
| | 1555 A>G | 5 |
| | 1226 G>A | 3 |
| | 2027 T>A | 1 |
| | 2168 A>G | 1 |
| | IVS7−2 A>G | 8 |
| Mongolian | 235 delC | 1 |
| | 299_300 delAT | 1 |

each; 3 cases detected at the c.1229 C>T locus, with a detection rate of 0.13%; 2 cases detected at the IVS15+5 G>A locus, with a detection rate of 0.09%. Mitochondrial 12S rRNA gene: 24 cases detected at the c.1555 A>G locus, with a detection rate of 1.03%; 1 case detected at the c.1494 C>T locus, with a detection rate of 0.04%.

### 3.7.2.3.3. GJB3 gene

1 case detected at the c.538 C>T locus, with a detection rate of 0.04%.

### 3.7.3. High-altitude areas (2,500 m < altitude ≤ 4,500 m). 3.7.3.1. Basic distribution

There are 742 newborns in this range, accounting for 24.20% of the total number of screened individuals (742/3,066). A total of 33 newborns carrying deafness mutant genes were detected in high-altitude areas, with an overall detection rate of 4.45%. Among them, there are 381 male infants and 361 female infants.

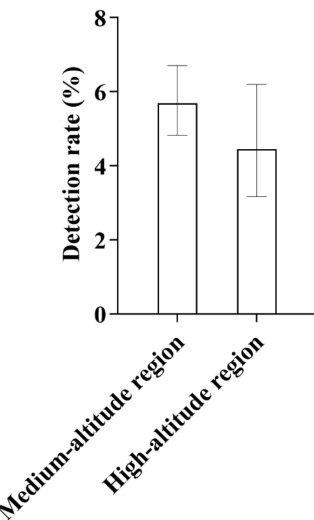

**Fig 22. Comparison of Detection Rates of Deafness-Susceptible Gene Mutations Between Medium- and High-Altitude Areas: As shown in the figure, the detection rate of deafness-susceptible gene mutations in high-altitude areas (4.45%) was lower than that in medium-altitude areas (5.69%).**

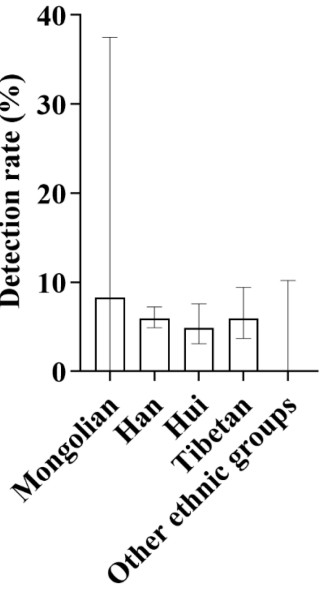

**Fig 23. Distribution of Detection Rates of Deafness-Susceptible Genes Among Newborns of Different Ethnic Groups in Medium-Altitude Areas: As shown in the figure, the Mongolian population exhibited the highest detection rate of deafness-susceptible genes (8.33%), followed by the Han population (5.96%).** The Tibetan population showed the lowest detection rate (4.43%).

### 3.7.3.2. Distribution and Detection Status of Ethnic Groups in High-Altitude Areas

386 Han nationality newborns, 20 cases detected, with a detection rate of 5.18%; 249 Tibetan nationality newborns, 7 cases detected, with a detection rate of 2.81%; 88 Hui nationality newborns, 5 cases detected, with a detection rate of 5.68%; 12 Mongolian nationality newborns, 1 case detected, with a detection rate of 8.33%; A total of 7 newborns from the other 4 ethnic groups (3 Tu, 2 Kazak, 1 Manchu, and 1 Miao), with no deafness mutant genes detected (Fig 25).

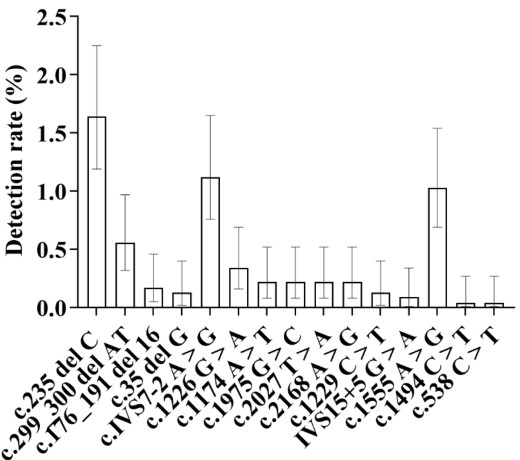

**Fig 24. The distribution of mutation rates at different loci of the GJB2 gene in medium-altitude areas is shown in the figure.** The c.235delC locus exhibited the highest mutation rate, with a detection rate of 1.64% (95%), followed by the **c.**IVS7-2A>G locus, with a detection rate of 1.12% (95%), and the **c.**1555A>G locus, with a detection rate of 1.03% (95%).

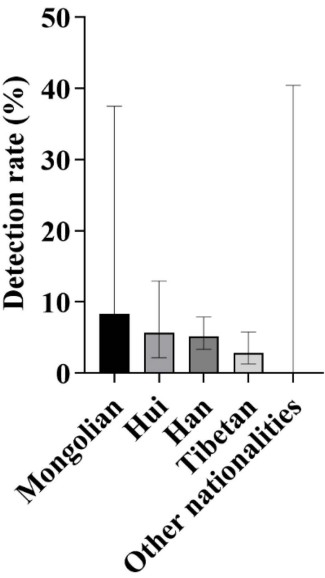

**Fig 25. Distribution of Detection Rates of Deafness-Susceptibility Genes Among Different Ethnic Groups in High-Altitude Areas: As shown in the figure, the Mongolian newborns exhibited the highest detection rate of deafness-susceptibility genes in high-altitude regions, reaching 8.33% (95%).** The Hui population ranked second, with a detection rate of 5.68%, which was higher than that of the Han population (5.18%). The Tibetan population showed the lowest detection rate, at only 2.81% (95%). No deafness-susceptibility genes were detected among the other ethnic groups.

### 3.7.3.3. Details of detection of various genes and loci in high-altitude areas (Fig 26)
### 3.7.3.3.1. GJB2 gene

11 cases detected at the c.235 del C locus, with a detection rate of 1.48%, which is the highest detection rate locus among newborns in high-altitude areas; 3 cases detected at the c.299_300 del AT locus, with a detection rate of 0.40%; 1 case detected at the c.35 del G locus, with a detection rate of 0.13%; No detection at the c.176_191 del 16 locus.

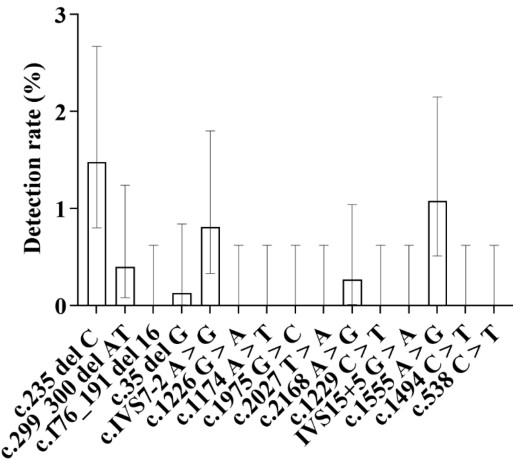

**Fig 26. Distribution of Mutation Detection Rates at Different Gene Loci in High-Altitude Areas.**

#### 3.7.3.3.2. SLC26A4 gene

6 cases detected at the c.IVS7−2 A>G locus, with a detection rate of 0.81%; 2 cases detected at the c.2168 A>G locus, with a detection rate of 0.27%; No detection at the six loci of c.1174 A>T, c.1226 G>A, c.1975 G>C, c.2027 T>A, c.1229 C>T, and IVS15+5 G>A.

#### 3.7.3.3.3. Mitochondrial 12S rRNA gene

8 cases detected at the c.1555 A>G locus, with a detection rate of 1.08%, which is the second highest detection rate locus after the c.235 del C locus; No detection at the c.1494 C>T locus.

#### 3.7.3.3.4. GJB3 gene: No detection.

As shown in the figure, the c.235delC locus exhibited the highest detection rate, at 1.48%. This was followed by the c.1555A>G locus, with a detection rate of 1.08%. The c.IVS7-2A>G locus, which previously ranked second, dropped to third place, with a detection rate of 0.81%.

## 4. Discussion

Internationally, relatively few newborn hearing loss gene screening efforts have been carried out in high-altitude areas (i.e., areas above 2,500 meters above sea level). In China, relevant studies are also mainly based on data from plain areas, with a lack of research data from plateau areas.

In terms of the order and types of hearing loss gene mutations, there is a specific sequence internationally. In developed countries, over 50% of prelingual deafness is caused by monogenic defects [9]. Among them, the GJB2 gene (which encodes connexin 26 and is also known as DFNB1) is a major pathogenic factor for prelingual deafness in multiple populations—this has been confirmed by linkage studies on deafness families in New Zealand, Australia, Italy, and Spain [10–16].

From the perspective of specific populations, the dominant role of GJB2 gene mutations and the diversity of mutation sites are particularly prominent: in Caucasian populations, 20%−50% of congenital deafness cases are associated with GJB2 mutations, 70% of which are 35delG mutations [12,17,18]; the average carrier rate of GJB2 35delG mutation in populations from Southern Europe is 2.8%, compared with 1.3% in populations from Central and Northern Europe [19]; in African populations, the main GJB2 mutation site is R143W, which is associated with both phenylketonuria (PKU) and hearing loss [20]; in deafness cases in Japan, GJB2 gene mutations also dominate, with the 235delC mutation being the most common [21]. Similarly, in this study, the locus with the highest detection rate of GJB2 gene mutations is also 235delC rather than 35delG.

Beyond GJB2, mutations in other genes also play important roles in the etiology of deafness: in Caucasian populations, at least 5% of postlingual non-syndromic hearing impairment is caused by known mitochondrial DNA (mtDNA) mutations, which are the most common etiology second only to the GJB2 35delG mutation [22]; in Japanese populations, the SLC26A4 gene mutation follows the deafness-susceptible GJB2 gene. In contrast, GJB3 gene mutations are relatively rarely detected [23].

Overall, in deaf populations such as those in Caucasian regions, Europe, America, and Japan, the trend of "GJB2 gene mutations being dominant, followed by SLC26A4 or mitochondrial gene mutations, with GJB3 gene mutations being rare" is observed, which is basically consistent with the overall situation in China, with only differences in specific mutation sites.

In China, the order from highest to lowest is GJB2 (3.10%), SLC26A4 (2.07%), Mt-12SrRNA (0.37%), and GJB3 (0.28%) (See Table 3 for details). Based on this, China currently takes these 4 deafness-susceptible genes (GJB2, SLC26A4, mitochondrial 12SrRNA, and GJB3) and their mutation sites as the main screening targets.

At present, deafness gene screening is actively carried out in various regions of China. Among them, the screening of four deafness susceptibility genes, namely GJB2 gene, SLC26A4 gene, mitochondrial 12SrRNA gene and GJB3 gene and their mutation sites is the most common. Among these genes, the detection rate of GJB2 gene mutation is the highest [24–26]; followed by SLC26A4 gene mutation. SLC26A4 gene mutation is closely related to the pathogenesis of large vestibular aqueduct syndrome in children [27]. Some children with mutations in the SLC26A4 gene do not have hearing loss at birth, and their hearing tests are usually normal. However, they are often induced by factors such as infection, fever and trauma, leading to delayed hearing loss [28,29]. The functional proteins expressed by the GJB2 gene and the SLC26A4 gene are Cx26 protein and Pendrin protein respectively. The functions of these two proteins are closely related to the maintenance of ion homeostasis in the inner ear. Once the GJB2 gene and the SLC26A4 gene mutate, the ion homeostasis of the inner ear will be disrupted [30,31]. At this time, the $K^+$ circulation is blocked and the lymphatic fluid reflux is hindered, which causes obstacles to the depolarization or hyperpolarization of the inner ear hair cells, and further contribute to a weakened ability of the hair cells to sense the mechanical waves of the endolymphatic fluid, or even an inability to sense it. Sound cannot be sensed normally or the sensing ability is greatly reduced, eventually resulting in hearing loss or even total deafness.

Interestingly, in this study, the detection rate of the c.IVS7−2 A > G locus in the SLC26A4 gene showed significant differences: the detection rate of this locus among newborns in high-altitude areas was 0.81%, lower than the 1.12% in medium-altitude areas; in the Tibetan population, its detection rate was 0.37%, much lower than that in the Hui nationality (1.68%) and the Han nationality (1.11%). Given that the Tibetan population resides more concentratedly in high-altitude areas compared with other ethnic groups, this suggests that the high-altitude environment may have a selective regulatory effect on the deafness gene mutation locus c.IVS7−2 A > G in the Tibetan population-this is not only a self-regulatory strategy formed by the Tibetan population in the long-term process of adapting to the high-altitude environment, but also explains why the overall detection rate of deafness genes in high-altitude areas is lower than that in medium-altitude areas.

In addition, it is noteworthy that the detection rate of the SLC26A4 gene among Hui newborns is the highest. Since this gene mutation is closely related to the onset of delayed-onset large vestibular aqueduct syndrome in children, carrying out targeted health education for the Hui population can greatly reduce the risk of such diseases.

On the contrary, the detection rate of deafness-susceptible genes among the Mongolian and Hui populations living in high-altitude areas is higher than that in low-altitude areas, which indicates that to a certain extent, the Mongolian and Hui nationalities have not formed a good adaptation to the high-altitude environment like the Tibetan population who have long settled in high-altitude areas.

Mutations in the mitochondrial 12SrRNA gene are closely associated with deafness induced by aminoglycoside drugs [32,33]. As a typical representative of maternally inherited genes, the genetic characteristics of the mitochondrial 12SrRNA deafness gene are directly related to its intracellular localization and genetic mechanism: since mitochondria are present

**Table 3. Screening Data of Neonatal Deafness Genes in Various Provinces of China.**

| Province | Altitude(m) | Mutation rate of gene(%) | | | | |
| --- | --- | --- | --- | --- | --- | --- |
| | | GJB2 | SLC26A4 | Mt-12SrRNA | GJB3 | Total |
| Tianjin | 3 | 4.80 | 4.50 | 0.20 | 0.40 | 5.53 |
| Jiangsu | 15 | 2.96 | 1.48 | 0.29 | 0.22 | 5.59 |
| Shandong | 39 | 2.59 | 2.34 | 0.44 | 0.39 | 5.66 |
| Beijing | 43.4 | 9.10 | 2.12 | 0.18 | 0.36 | 11.64 |
| Guangdong | 90 | 1.74 | 1.39 | 0.51 | 0.32 | 3.68 |
| Henan | 100 | 3.44 | 1.88 | 0.29 | 0.71 | 5.92 |
| Anhui | 119 | 3.20 | 1.46 | 0.36 | 0.36 | 5.29 |
| Hainan | 120 | 2.27 | 1.49 | 0.27 | 0.14 | 18.47 |
| Jiangxi | 246 | 2.60 | 0.98 | 0.37 | 0.10 | 4.05 |
| Hunan | 350 | 2.22 | 1.15 | 0.39 | 0.20 | 4.23 |
| Liaoning | 400 | 1.57 | 0.94 | 0.13 | 0.18 | 5.33 |
| Zhejiang | 400 | 2.75 | 1.55 | 0.29 | 0.27 | 4.89 |
| Jilin | 450 | – | – | – | – | – |
| Fujian | 500 | 1.41 | 1.05 | 0.28 | 0.19 | 2.95 |
| Heilongjiang | 500 | 11.74 | 11.29 | 0.46 | 0 | 23.48 |
| Taiwan | 778 | 18.88 | 059 | 0.10 | – | 19.57 |
| Guangxi Zhuang Autonomous Region | 800 | 1.38 | 0.42 | 0.03 | 0.35 | 2.98 |
| Shanxi | 1000 | 1.69 | 2.13 | 0.37 | 0.33 | 4.15 |
| Shaanxi | 1000 | 2.06 | 1.91 | 0.18 | 0.06 | 4.72 |
| Xinjiang Uygur Autonomous Region | 1000 | 1.74 | 1.02 | 0.34 | – | 3.04 |
| Inner Mongolia Autonomous Region | 1100 | 1.83 | 1.79 | 0.81 | 0.38 | 4.31 |
| Ningxia Hui Autonomous Region | 1100 | 2.11 | 1.98 | 0.23 | 0.25 | 4.13 |
| Guizhou | 1100 | 2.28 | 1.76 | 0.45 | 0.43 | 4.92 |
| Hebei | 1500 | 6.34 | 3.57 | 0.77 | 0.56 | 10.43 |
| Gansu | 1500 | 1.40 | 1.58 | 0.49 | 0.27 | 3.80 |
| Hubei | 1800 | 2.29 | 1.15 | 0.49 | 0.07 | 3.70 |
| Sichuan | 1849 | – | – | – | – | – |
| Yunnan | 2000 | 2.10 | 0.80 | 0.41 | 0.28 | 3.68 |
| Tibet Autonomous Region | 4000 | – | – | – | – | – |
| Qinghai | 4058 | – | – | – | – | – |
| Special Administrative Region | – | – | – | – | – | – |
| **Average value** | – | **3.10** | **2.07** | **0.37** | **0.28** | **6.67** |

in the cytoplasm, and the cytoplasm (containing mitochondria) of offspring is mainly provided by the mother during inheritance, this gene exhibits strict maternal inheritance characteristics-if this gene is detected in a newborn, all maternal members of the family are usually carriers.

Notably, the overall detection rate of the mitochondrial 12SrRNA gene in this study reached 1.08%, significantly higher than the national average level (0.37%), and this difference is particularly prominent at its c.1555 A>G locus. Further analysis of ethnic distribution shows that the detection rate of the c.1555 A>G locus in the Tibetan population is 1.50%, which is significantly higher than that in the Hui nationality (1.05%) and the Han nationality (0.96%). Based on the above findings, in the Tibetan and Hui populations in high-altitude areas, implementing early education and medication guidance for children detected with this gene mutation and all their maternal family members can significantly reduce the risk of irreversible deafness caused by exposure to aminoglycoside drugs. This intervention measure undoubtedly has important preventive value.

The GJB3 gene can cause autosomal recessive or dominant hereditary non-syndromic deafness [34]. Its detection rate is lower than that of the above three deafness susceptibility genes. This gene is located at chromosome 1p34.3 and encodes Cx31 protein. It participates in the construction of intercellular ion channels together with Cx26 protein and Cx30 protein to maintain ion homeostasis in the inner ear microenvironment [35].

From the perspective of traditional drug treatment, the treatment of this type of deafness is rather difficult. Therefore, in the current field of deafness treatment, gene therapy for deafness has received much attention. Especially in 2024, the dual-vector gene replacement therapy (AAV1-hOTOF drug) for adeno-associated virus (AAV) by Shu et al [36] achieved remarkable effect in the treatment of patients with OTOF deafness. For patients with hereditary deafness, especially children, this is undoubtedly an exciting research! In addition, stem cell regeneration therapy for inner ear hair cells is also a current research hotspot. This is a research method based on results or established facts. If hair cell damage can be replaced or repaired through the regeneration of inner ear hair cell stem cells or the redifferentiation of supporting cells, it may be possible to treat deafness caused by various reasons without being limited to a specific type of deafness.

To further illustrate the relationship between the data of this study and the data of deafness gene screening in other regions across the country more intuitively, we have listed the results of deafness gene screening that have been carried out in China in recent years by province respectively [10,37–88]. As shown in Table 3.

Explanation: Most of the data in the table are the means of the data from different regions within a certain province in different literature. China has a total of 23 provinces and 5 autonomous regions. It should be noted that the data in the above table only represent the information that was retrieved and met the inclusion criteria of this study within a specific time frame.

Through the analysis of the above table, we can quickly understand the current situation of screening for neonatal deafness susceptibility genes in various provinces in China. It is noteworthy that, the average detection rates of deafness genes GJB2, SLC26A4, Mt-12SrRNA and GJB3 in China are 3.10%, 2.07%, 0.37% and 0.28% respectively, and the overall detection rate of deafness susceptibility genes is around 6.67%. In some regions of China, such as Beijing, Hainan, Heilongjiang, Taiwan and Hebei, the detection rates of deafness susceptibility genes show relatively high levels. Through in-depth exploration of the reasons behind this, it can be known that it is mainly closely related to multiple factors, including the differences in the types and numbers of gene loci targeted during the detection process, as well as the diversity of ethnic compositions in different regions. These factors are intertwined and work together, resulting in the fact that the detection situations of deafness susceptibility genes in the above-mentioned regions are different from those in other regions, thus showing a relatively high detection rate trend in the overall data. For example, in the research conducted by WU et al. [46] in Taiwan, the detection of the p.V37I/wt site was included, which led to a relatively high detection rate of the GJB2 gene. If the detection data of the p.V37I/wt site were excluded, the detection rate of common deafness susceptibility genes in the Taiwan region would be basically on a par with the national level. In this table, the research subjects used in the neonatal deafness gene screening in Heilongjiang region were newborns who did not pass the hearing screening [45]. Since these newborns themselves have a relatively high risk of hearing abnormalities, the detection rate is correspondingly higher when conducting deafness gene testing. It should be noted that this data is used because other screening data resources targeting the neonatal population have not been obtained temporarily in Heilongjiang region at present. Therefore, it can only be presented and analyzed as a reference. Generally speaking, to some extent, the more types of mutation sites of deafness susceptibility genes are detected, the relatively higher the detection rate of deafness genes will be.

Furthermore, through the analysis of the above data, it can be clearly found that there is a significant lack of research on the screening data of neonatal deafness genes in regions such as Jilin, Xinjiang, Sichuan, Tibet, Hong Kong and Macau at present. Meanwhile, the deafness gene screening data covered in the table are mainly sourced from low-altitude areas, and the screening work for neonatal deafness genes in high-altitude areas (that is, areas with an altitude higher than 2,500 meters) is relatively scarce. In view of this, whether altitude will have an impact on the detection rate of

deafness genes urgently requires further in-depth research and the strong support of more detailed data, so as to have a more comprehensive and accurate understanding of the situation and patterns of neonatal deafness gene screening in different regional environments.

This study found that the detection rate of deafness-susceptible genes in newborns from high-altitude areas was significantly lower than that in low-altitude areas. This result seems to contradict existing research conclusions-previous studies have shown that human auditory and balance functions deteriorate in the hypobaric and hypoxic plateau environment [89–93].

In-depth analysis reveals that the core of this apparent contradiction lies in the fundamental differences in the characteristics of the study populations: The subjects of this study are long-term residents of high-altitude areas, who have developed a good tolerance mechanism to the plateau environment through long-term adaptation. Specifically, the detection rate of the high-frequency mutation site IVS7−2 A>G of the SLC26A4 gene in the Tibetan population in high-altitude areas is extremely low, and no mutations in loci such as 176_191 del 16, 1174 A>T, 1226 G>A, 1229 C>T, 1975 G>C, or in the GJB3 gene have been detected in these areas. These adaptive characteristics at the genetic level may be important reasons for the lower detection rate of deafness-susceptible genes.

In contrast, the subjects in the aforementioned literatures are either groups that have acutely entered high-altitude areas or are in scenarios simulating acute plateau exposure. Such populations have not yet established an adaptation mechanism to the hypobaric and hypoxic environment, and their symptoms such as dizziness, tinnitus, and hearing loss are mainly related to physiological stress responses caused by acute hypoxia, such as changes at the tissue level and increased red blood cells [94,95].

It can be seen that the two sets of conclusions belong to studies under different physiological states and do not constitute a real contradiction. Instead, they reveal the association between the plateau environment and auditory function as well as related genetic characteristics from different perspectives.

In terms of the correlation between ethnic differences and the detection rate of deafness genes, the current research situation is rather complex. The existing research results have not reached a unified conclusion. Some research results strongly confirm that ethnic differences have a significant and non-negligible impact on the detection rate of deafness genes [96–100]. However, research conclusions that are quite different and contradictory are also common [101]. Such a divergence in research conclusions deeply reflects the urgency and necessity of conducting further in-depth exploration. There is an urgent need to collect more abundant and diverse data under a more rigorous and unified standard framework for accurate verification. Why is this so? Because in different geographical regions and among various ethnic groups, the mutation sites of neonatal deafness susceptibility genes show extremely significant diversity and differentiation characteristics. This undoubtedly adds many unpredictable uncertainties and severe challenges to deeply exploring the inherent and hidden correlation between ethnic differences and the detection rate of deafness genes. At the roundtable forum on "Clinical Diagnosis and Treatment Research of Hereditary Deafness" held in 2023 [102], both Professor Zuhong H and Professor Limin S emphasized the great importance and far-reaching significance of gene screening work in ethnic minority areas.

This study selected the Xining area of Qinghai Province (a high-altitude hypoxic region) as the research site. This area has a rich and diverse ethnic composition, covering multiple ethnic groups such as Tibetan, Hui, Tu, Salar, Mongolian, and Han. In the screening of hereditary deafness-susceptible genes among newborns in this region, the study found that the detection rate of SLC26A4 gene mutations was the highest. This result shows an obvious difference from the distribution characteristic in low-altitude areas, where GJB2 gene mutations are dominant-this difference may be related to the extremely high proportion of the Han population in low-altitude areas, thus highlighting the unique distribution pattern of deafness-susceptible genes in high-altitude multi-ethnic regions.

This study selected Xining area of Qinghai Province (a high-altitude hypoxic region) as the research site. This area has a rich and diverse ethnic composition, covering multiple ethnic groups such as Tibetan, Hui, Tu, Salar, Mongolian, and

Han. In this study, the total detection rate of 15 loci in 4 genes was 5.38%, which was lower than the national average detection level (6.67%). This difference may be related to the geographical and demographic characteristics of Qinghai Province as a medium-high altitude plateau area and a multi-ethnic settlement.

Among the 4 genes screened, the detection rate of GJB2 gene was the highest (2.38%), with the 235 delC locus having the highest mutation rate, and the overall detection level was lower than the national average (3.10%); the detection rate of GJB3 gene mutation was 0.03%, also lower than the national average; while the detection rates of SLC26A4 gene and mitochondrial 12SrRNA gene reached 2.19% and 1.08% respectively, both higher than the national average levels (2.07% and 0.37% respectively).

Data analysis showed that in medium-high altitude areas, especially high-altitude regions, the detection rate of deafness mutant genes showed a decreasing trend, and some gene loci were even not detected, suggesting that the high-altitude environment may have a selective effect on deafness gene mutations. This result shows an obvious difference from the distribution characteristics of deafness gene screening in low-altitude areas -this difference may be related to the extremely high proportion of the Han population in low-altitude areas, thus highlighting the unique distribution pattern of deafness-susceptible genes in high-altitude multi-ethnic regions.

Interestingly, in this study, we found 2 cases of neonatal external auditory canal atresia and 1 case of auricular malformation, but their genetic test results were all negative. This phenomenon indicates that in addition to genetic screening, it is still necessary to supplement and improve the screening indicators related to structural malformations to reduce the incidence of such birth defects.

This study still has certain limitations: First, the detection rate of deafness-susceptible genes in Mongolian newborns in this study is significantly higher than that in newborns of other ethnic groups, and this phenomenon may be associated with the small sample size of Mongolian newborns; Second, the situation where no positive cases were detected in other ethnic minorities is also likely limited by insufficient sample size; Third, this study only screened specific loci and variants of 15 loci across 4 deafness-related genes, with the selection criteria based on the clinically common deafness gene screening panel and the high-frequency pathogenic variants reported in the local population. However, this targeted screening may miss other potential deafness-causing loci or rare variants, which may lead to an underestimation or bias in the detection rate of deafness-susceptible genes; Fourth, the frequency data reported in this study are only based on the selected mid-altitude and high-altitude areas, and no plain areas were included as controls (only mid-altitude areas were used as controls for high-altitude areas). This makes it impossible to rule out the influence of geographical factors other than altitude (such as environmental conditions, living habits, and dietary structure) on the detection rate of deafness-susceptible genes, thus limiting the generalizability of the research results to a certain extent. In subsequent studies, it is necessary to further expand the sample size of newborns from various ethnic minorities, optimize the deafness gene screening panel by incorporating more potential pathogenic loci based on comprehensive genetic and epidemiological data, and include plain areas as controls to clarify the true causes of the aforementioned differences in detection rates, the absence of positive cases, and the impact of altitude and other geographical factors on the distribution of deafness-susceptible genes.

In fact, one point that we cannot overlook regarding deafness is that in clinical practice, the proportion of acquired deafness is even larger. Since the factors that induce acquired deafness are numerous and complex, a traditional, single, and limited study of a certain aspect is hardly sufficient to meet the requirements. Perhaps in the future, multi-omics and systematic studies will likely demonstrate even greater significance. This is undoubtedly the focus and also the difficulty of the next step of research.

## 5. Conclusion

1. Overall level: The overall carriage rate of deafness susceptibility genes in neonates from Xining, Qinghai is significantly lower than the national average for newborns (6.67%). Within the medium and high altitude regions of Qinghai Province, compared with the medium-altitude areas (altitude > 1,500 meters and ≤ 2,500 meters), the carriage rate of

deafness susceptibility genes in neonates from high-altitude areas (altitude > 2,500 meters and ≤ 4,500 meters) shows a downward trend, which suggests that within the medium and high altitude range of the province, the detection rate of deafness susceptibility genes decreases with the increase of altitude.

2. Gene level: Among GJB2 gene mutations, the 235 delC gene mutation remains the most dominant type, and the detection rates of both GJB2 and GJB3 genes are lower than the national level. Notably, the detection rates of mitochondrial 12S rRNA gene mutations and SLC26A4 gene mutations in neonates from this region are both higher than the national average, with the detection rate of mitochondrial 12S rRNA gene mutations being more significantly higher. Since deafness caused by these two types of gene mutations can be effectively reduced in terms of onset risk through health education, conducting genetic testing targeting these specific genes has important clinical significance and public health value.

3. Ethnic background level: The detection rate of deafness susceptibility genes in Tibetan neonates is low, especially for the IVS7−2 A>G mutation site, and this phenomenon is more prominent in high-altitude areas. On the contrary, the detection rate of deafness susceptibility genes in Mongolian and Hui neonates in high-altitude areas shows an increasing trend.

## Supporting information

**S1 File. Research Plan.** English version.
(PDF)

**S1 Table. Table 4 Stratified Comparison Chart of Detection Rate of Deafness Susceptibility Genes.**
(DOCX)

## Acknowledgments

The authors acknowledge the valuable contributions and strong support provided by the Department of Otorhinolaryngology, Department of Pediatrics, and Department of Neonatal Intensive Care Medicine of the Affiliated Hospital of Qinghai University to this research.

## Author contributions

**Conceptualization:** Yi Wang, Yazhen Wu, Yanyan Ma.

**Data curation:** Yi Wang, BenHong Ren, Wenyuan Gan, Qingping Zhang.

**Formal analysis:** Yi Wang, Benhong Ren, Qingping Zhang, Yanyan Ma.

**Funding acquisition:** Yi Wang, Yanyan Ma.

**Investigation:** Yi Wang, BenHong Ren, Yazhen Wu, Wenyuan Gan, Qingping Zhang, Dandan Yang, Wenjun Cao, Xiaoli Zhang, Ying Zhang, Bin Guo, Yongxia Tie, Jiannan Liu, Guilan Cai, Yanyan Ma.

**Methodology:** Yi Wang, Yazhen Wu, Dandan Yang, Yuan Jiang, Yanyan Ma.

**Project administration:** Yi Wang, Yazhen Wu, Dandan Yang, Yanyan Ma.

**Resources:** Yi Wang, Yazhen Wu, Dandan Yang, Wenjun Cao, Xiaoli Zhang, Ying Zhang, Bin Guo, Yongxia Tie, Jiannan Liu, Guilan Cai, Yanyan Ma.

**Software:** BenHong Ren.

**Supervision:** Yi Wang, Yazhen Wu, Yuan Jiang.

**Validation:** BenHong Ren, Yuan Jiang.

**Visualization:** BenHong Ren.

**Writing – original draft:** BenHong Ren.

**Writing – review & editing:** Yi Wang.

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
