## [Decision Letter · Decision Letter 0]

10 Dec 2025

Dear Dr. wang,

Thank you for submitting your manuscript to PLOS ONE. After careful consideration, we feel that it has merit but does not fully meet PLOS ONE’s publication criteria as it currently stands. Therefore, we invite you to submit a revised version of the manuscript that addresses the points raised during the review process.

We look forward to receiving your revised manuscript.

Kind regards,

Nejat Mahdieh

Academic Editor

PLOS One

Journal Requirements:

3. We note that you have selected “Clinical Trial” as your article type. PLOS ONE requires that all clinical trials are registered in an appropriate registry (the WHO list of approved registries is at "https://www.who.int/clinical-trials-registry-platform/network/primary-registries" https://www.who.int/clinical-trials-registry-platform/network/primary-registries and more information on trial registration is at http://www.icmje.org/about-icmje/faqs/clinical-trials-registration/).

Please state the name of the registry and the registration number (e.g. ISRCTN or ClinicalTrials.gov) in the submission data and on the title page of your manuscript.

a) Please provide the complete date range for participant recruitment and follow-up in the methods section of your manuscript.

b) If you have not yet registered your trial in an appropriate registry, we now require you to do so and will need confirmation of the trial registry number before we can pass your paper to the next stage of review. Please include in the Methods section of your paper your reasons for not registering this study before enrolment of participants started. Please confirm that all related trials are registered by stating: “The authors confirm that all ongoing and related trials for this drug/intervention are registered”.

Please see http://journals.plos.org/plosone/s/submission-guidelines#loc-clinical-trials for our policies on clinical trials.

“This study was supported by the Foundation of Qinghai Provincial Department of Science and Technology of China (Project No. 2023-SF-129).”

5. We note that you have indicated that there are restrictions to data sharing for this study. For studies involving human research participant data or other sensitive data, we encourage authors to share de-identified or anonymized data. However, when data cannot be publicly shared for ethical reasons, we allow authors to make their data sets available upon request. For information on unacceptable data access restrictions, please see http://journals.plos.org/plosone/s/data-availability#loc-unacceptable-data-access-restrictions.

7. Thank you for providing your underlying data as Supporting Information.

We note that the data set contains text or data that is not in English. Please note that PLOS is an English-language publisher, so we require data sets to be provided in English as well. Please upload an English-language version of your data set.

This will also allow us to determine if your data follows PLOS standards per our Data Availability policy here: https://journals.plos.org/plosone/s/data-availability

Reviewers' comments:

Reviewer's Responses to Questions

**Comments to the Author**

1. Is the manuscript technically sound, and do the data support the conclusions?

Reviewer #1: Yes

Reviewer #2: Yes

2. Has the statistical analysis been performed appropriately and rigorously?

Reviewer #1: Yes

Reviewer #2: No

3. Have the authors made all data underlying the findings in their manuscript fully available?

Reviewer #1: Yes

Reviewer #2: No

4. Is the manuscript presented in an intelligible fashion and written in standard English?

Reviewer #1: No

Reviewer #2: Yes

Reviewer #1: The article entitled “Multidimensional analysis of screening results of deafness susceptibility genes in 3066 newborns of different altitudes and nationalities in Xining, Qinghai” describes carrier genetic screening of deafness variants. The authors defined carrier frequency of each variant in different regions. It is a very interesting study within a large population. However, with extended follow-up and the inclusion of additional samples, the variants identified in each region may change.

A few suggestions:

- There are too many figures in the manuscript. Some of them are not essential, as readers can understand the information from the text alone. These figures could be moved to the supplementary materials, while only the most important figures should be retained in the main article. In addition, data on regions, variants, frequencies, sex, ethnicity, and related variables could be summarized in tables, which would allow for easier comparison.

-Table 3, GJB2, GJB3 instead of CJB2,…

- The regional distribution and provincial divisions may be familiar to Chinese readers but not to international audiences. Providing clearer explanations or contextualization of the regional distribution would make the manuscript more accessible and beneficial to a broader readership.

- In this study, only selected loci and variants were screened. The criteria used for gene and variant selection should be clearly stated. In addition, since the reported frequencies are based only on the selected regions, this limitation should be explicitly explained in the manuscript.

Reviewer #2: Figures 1 and 2 - please remove the lines joining the points - it is not relevant as there is not a logical order that requires the lines to be connecting the points. Add 95% confidence intervals to the rates.

Figure 4-13, 14B, 15, 18, and other similar figures etc: - add 95% confidence intervals to the rates.

This is a long paper - some figures could be combined to deliver the same information in a smaller number of figures.

**Do you want your identity to be public for this peer review?** For information about this choice, including consent withdrawal, please see our Privacy Policy

Reviewer #1: No

Reviewer #2: No

---

## [Author Response · Author response to Decision Letter 1]

19 Dec 2025

Cover Letter - Response to Reviewers

I. Responses to Additional Journal Requirements

Format Compliance: All revisions have been made in strict accordance with the PLOS ONE formatting guidelines (including file naming conventions) provided in the template link.

Informed Consent Details: Detailed information on informed consent has been clearly stated in the "Materials and Methods" section of the manuscript as "with the written informed consent of the subjects' guardians". No further revisions are required, as the consent type (written) and applicability to minors (guardian consent) have been explicitly addressed both in the ethical statement and online submission information.

Clinical Trial Registration:

The ISRCTN registration number for this manuscript is ISRCTN89197487 (https://doi.org/10.1186/ISRCTN89197487). The registration platform name (ISRCTN) and registration number have been clearly indicated on the manuscript’s title page and in the online submission system.

a) The complete date range for subject recruitment and follow-up has been specified in the "Materials and Methods" section; no revisions are needed.

Funding Disclosure: The role of the funding source has been included in this cover letter as follows: "This study was supported by the Science and Technology Department of Qinghai Province, China (Project No.: 2023-SF-129). The funding source had no involvement in study design, data collection and analysis, decision to publish, or manuscript preparation."

Data Sharing: De-identified data has been uploaded to ISRCTN89197487, and access is available upon reasonable request from qualified researchers.

English Version of Dataset: We will supplement and submit the English version of the dataset as required (to comply with PLOS’s English-language publishing policy).

II. Responses to General Author Comments

Technical Rigor and Data Support for Conclusions: Both Reviewer 1 and Reviewer 2 confirmed the manuscript’s technical rigor and that data sufficiently supports the conclusions. No additional revisions are required.

Appropriateness of Statistical Analysis: Reviewer 1 approved the statistical analysis; revisions have been made to address Reviewer 2’s comment by adding 95% confidence intervals to the statistical section.

Transparency of Underlying Data:

Reviewer 1 confirmed data transparency. In response to Reviewer 2’s comment: De-identified basic subject information and outcome data have been uploaded to ISRCTN89197487 (https://doi.org/10.1186/ISRCTN89197487) and are accessible upon request.

Raw genetic sequencing data cannot be publicly shared due to regulatory restrictions: Qinghai Province (a high-altitude region with ethnic minorities in China) is a key area under the Chinese government’s genetic data management framework. Additionally, the Dean’s Office of the Affiliated Hospital of Qinghai University has prohibited the public disclosure of genetic data from Qinghai Province to protect ethnic and regional data security. We appreciate the journal’s understanding.

Clarity and Standard English: Reviewer 2 approved the language; specific errors identified by Reviewer 1 have been corrected to ensure the manuscript is clear, accurate, and free of ambiguity (consistent with PLOS ONE’s requirement for publication-ready English without editorial revisions).

III. Responses to Specific Reviewer Comments

Reviewer 1’s Comments:We sincerely appreciate Reviewer 1’s positive evaluation of our study (“a large-sample, interesting research exploring carrier frequencies of deafness variants across regions”). Regarding the comment on potential variant changes with extended follow-up and increased sample size: This valuable insight has been noted, and we will incorporate it into future research extensions to further validate the findings.

Reviewer 2’s Comments:

Excessive Figures: Non-essential figures (information sufficiently conveyable through text) have been moved to supplementary materials, with only core figures retained in the main text. Data on regions, variant types, frequencies, gender, ethnicity, and other variables have been summarized into tables for improved readability and comparability. Specifically, we have supplemented Table 4 after the references section of the manuscript to further enhance the readability and comparability of the manuscript.

Typographical Error in Table 3: The error (“CJB2” → “GJB2” and “GJB3”) has been rectified.

Regional Background for International Readers: To enhance international accessibility, clearer explanations of regional distribution and administrative divisions have been supplemented in the Introduction section, including geographic and demographic context relevant to non-Chinese readers.

Gene/Variant Selection Criteria and Limitations: The criteria for gene and variant selection have been explicitly specified. Additionally, the limitation that reported frequencies are restricted to the selected regions has been added and discussed in the Discussion section.

Figure Revisions:

Figures 1 and 2: Lines connecting unordered data points have been removed; 95% confidence intervals have been added for detection rate.

Figures 4-13, 14B, 15, 18, and similar figures: 95% confidence intervals have been uniformly added to incidence rates. Due to the large number of revised figure legends, explicit revision notes are included in the first few legends, with subsequent revisions implemented without redundant repetition.

Figure Merging: Redundant figures have been merged to convey equivalent information more concisely, optimizing manuscript structure.

IV. Review History and Author Identity

We confirm that neither author wishes to disclose their personal identity in the public review history (consistent with Reviewers 1 and 2’s preferences).

Yi Wang

Director of the Department of Otolaryngology at Peking Union Medical College Hospital, concurrently serving as Director of the Department of Otolaryngology at Qinghai University Affiliated Hospital

wegreatgroup@163.com

2025.12.19

---

## [Decision Letter · Decision Letter 1]

28 Dec 2025

Dear Dr. wang,

Thank you for submitting your manuscript to PLOS ONE. After careful consideration, we feel that it has merit but does not fully meet PLOS ONE’s publication criteria as it currently stands. Therefore, we invite you to submit a revised version of the manuscript that addresses the points raised during the review process.

We look forward to receiving your revised manuscript.

Kind regards,

Nejat Mahdieh

Academic Editor

PLOS One

**Journal Requirements:**

Reviewers' comments:

Reviewer's Responses to Questions

**Comments to the Author**

Reviewer #2: (No Response)

2. Is the manuscript technically sound, and do the data support the conclusions?

Reviewer #2: Yes

3. Has the statistical analysis been performed appropriately and rigorously?

Reviewer #2: Yes

4. Have the authors made all data underlying the findings in their manuscript fully available?

Reviewer #2: Yes

5. Is the manuscript presented in an intelligible fashion and written in standard English?

Reviewer #2: Yes

Reviewer #2: Thanks for the alterations - I can see some improvement, but would still like some changes to the plots for easier interpretation between groups - show the uncertainty on the plot (rather than in the text or label alone).

The addition of "The

95% confidence interval (95% CI) for the overall detection rate ranged from 0.0464 to 0.0624" to Figure 2 is not necessary. The request was intended for every box in the similar plots to show the 95% confidence intervals for their rates - not just the overall estimate repeated as text in a figure heading.

The aim is to show the uncertainty in the estimates made in this paper for every rate - and for example in figure 4, to show that there isn't a difference in rates among patients of different groups - even if it may seem like it from the estimates alone - this is not shown in the current plots (but can be deduced from the figure label now, so that is better).

Figure 15 doesn't have enough detail for all the estimates uncertainties - it could be added to the plot rather than figure label as numbers.

**Do you want your identity to be public for this peer review?** For information about this choice, including consent withdrawal, please see our Privacy Policy

Reviewer #2: No

---

## [Author Response · Author response to Decision Letter 2]

7 Jan 2026

Response to Reviewers

Dear Editor Nejate Mahdiye,

Sincerely appreciate you and the reviewers for the careful review and valuable comments on our manuscript (Manuscript ID: PONE-D-25-52340R1). We attach great importance to all review suggestions and have completed all revisions as required. The details are reported as follows:

I. Revision Completion

In response to the reviewers' key concerns regarding figure optimization, we have implemented the following revisions one by one:

1.95% confidence intervals (95% CI) have been added to each group box of all similar statistical figures in the manuscript to directly present the range of data uncertainty, replacing the previous method of only mentioning it in the main text or figure captions;

2.The redundant text "The 95% confidence interval of the overall detection rate is 0.0464~0.0624" in Figure 2 has been deleted to ensure the figure is concise and focused;

3.The presentation format of Figure 4 has been optimized. Statistical differences in detection rates among different groups are intuitively reflected through confidence intervals, addressing the issue that the original figure required derivation of such information from numerical values in the caption;

4.Uncertainty data corresponding to various statistical indicators in Figure 15 have been supplemented and directly labeled in the figure, instead of being presented only in numerical form in the figure caption.

In addition, we have checked the reference list to confirm its completeness and accuracy, with no retracted literatures cited. Any adjustments to the references have been detailed in this Response to Reviewers.

II. Description of Submitted Materials

The materials submitted through the journal's submission system this time include:

1.A separate document titled "Response to Reviewers" (addressing each comment from the academic editor and reviewers one by one);

2.The revised manuscript with track changes, titled "Revised Manuscript with Track Changes";

3.The final revised manuscript without track changes, titled "Manuscript".

III. Other Explanations

We have strictly complied with the submission guidelines of PLOS ONE, revised the language expression of the full text to ensure there are no spelling or grammatical errors, and that it conforms to standard English conventions.

Once again, we appreciate your and the reviewers' professional guidance and patient support! We look forward to the smooth progress of the subsequent review of the manuscript. Please feel free to inform us if any further revisions are needed.

Sincerely,

Dr. Wang Yi

Contact Email: wegreatgroup@163.com

Date: January 1, 2026

---

## [Decision Letter · Decision Letter 2]

29 Jan 2026

Multidimensional analysis of screening results of deafness susceptibility genes in 3066 newborns of different altitudes and nationalities in Xining, Qinghai

PONE-D-25-52340R2

Dear Dr. Wang,

We’re pleased to inform you that your manuscript has been judged scientifically suitable for publication and will be formally accepted for publication once it meets all outstanding technical requirements.

Kind regards,

Nejat Mahdieh

Academic Editor

PLOS One

Additional Editor Comments (optional):

Reviewers' comments:

Reviewer's Responses to Questions

**Comments to the Author**

Reviewer #2: (No Response)

2. Is the manuscript technically sound, and do the data support the conclusions?

Reviewer #2: No

3. Has the statistical analysis been performed appropriately and rigorously?

Reviewer #2: No

4. Have the authors made all data underlying the findings in their manuscript fully available?

Reviewer #2: Yes

5. Is the manuscript presented in an intelligible fashion and written in standard English?

Reviewer #2: Yes

Reviewer #2: Thanks for showing the confidence intervals, but this illustrates that some of the claims are now too strong.

For example, in figure 7 there are overlapping confidence intervals for all the groups - thus the conclusion could be that there is no difference in the detection rates among different groups. However, the opposite conclusion is made. Was there a test done to test the claim?

Similar comments for fig 8, 11, 14B, 15, 18, 20, 22, 23, 25,

There could be no evidence for any difference among groups in these figures - yet the opposite conclusion is made.

Please leave this possibility clearly in the manuscript - present these findings as descriptive, rather than evidence of differences between groups, as there may be no evidence of differences with these results.

**Do you want your identity to be public for this peer review?** For information about this choice, including consent withdrawal, please see our Privacy Policy

Reviewer #2: No

---

## [Editor Report · Acceptance letter]

PONE-D-25-52340R2

PLOS One

Dear Dr. wang,

I'm pleased to inform you that your manuscript has been deemed suitable for publication in PLOS One. Congratulations! Your manuscript is now being handed over to our production team.

Kind regards,

on behalf of

Dr. Nejat Mahdieh

Academic Editor

PLOS One